# Primal-Dual Block Generalized Frank-Wolfe

Qi Lei[†*], Jiacheng Zhuo[†*], Constantine Caramanis[†], Inderjit S. Dhillon[†‡], and Alexandros G. Dimakis[†]

[†] UT Austin  [‡] Amazon
```
{leiqi@oden., jzhuo@, constantine@, inderjit@cs.,
            dimakis@austin.}utexas.edu
```

## Abstract

We propose a generalized variant of Frank-Wolfe algorithm for solving a class of sparse/low-rank optimization problems. Our formulation includes Elastic Net, regularized SVMs and phase retrieval as special cases. The proposed Primal-Dual Block Generalized Frank-Wolfe algorithm reduces the per-iteration cost while maintaining linear convergence rate. The per iteration cost of our method depends on the structural complexity of the solution (i.e. sparsity/low-rank) instead of the ambient dimension. We empirically show that our algorithm outperforms the state-of-the-art methods on (multi-class) classification tasks.

## 1 Introduction

We consider optimization problems of the form:

$$\min_{\boldsymbol{x} \in C} : \sum_i f_i(\boldsymbol{a}_i^\top \boldsymbol{x}) + g(\boldsymbol{x}),$$

directly motivated by regularized and constrained Empirical Risk Minimization (ERM). Particularly, we are interested in problems whose solution has special "simple" structure like low-rank or sparsity. The sparsity constraint applies to large-scale multiclass/multi-label classification, low-degree polynomial data mapping [5], random feature kernel machines [32], and Elastic Net [39]. Motivated by recent applications in low-rank multi-class SVM, phase retrieval, matrix completion, affine rank minimization and other problems (e.g., [9, 31, 2, 3]), we also consider settings where the constraint $\boldsymbol{x} \in C$ (e.g., trace norm ball) while convex, may be difficult to project onto. A wish-list for this class of problems would include an algorithm that (1) exploits the function finite-sum form and the simple structure of the solution, (2) achieves linear convergence for smooth and strongly convex problems, (3) does not pay a heavy price for the projection step.

We propose a Frank-Wolfe (FW) type method that attains these three goals. This does not come without challenges: Although it is currently well-appreciated that FW type algorithms avoid the cost of projection [14, 1], the benefits are limited to constraints that are hard to project onto, like the trace norm ball. For problems like phase retrieval and ERM for multi-label multi-class classification, the gradient computation requires large matrix multiplications. This dominates the per-iteration cost, and the existing FW type methods do not asymptotically reduce time complexity per iteration, even without paying the expensive projection step. Meanwhile, for simpler constraints like the $\ell_1$ norm ball or the simplex, it is unclear if FW can offer any benefits compared to other methods. Moreover, as is generally known, FW suffers from sub-linear convergence rate even for well-conditioned problems that enjoy strong convexity and smoothness.

**Our contributions.** In this paper we tackle the challenges by exploiting the special structure induced by the constraints and FW steps. We propose a generalized variant of FW that we call Primal-Dual Block Generalized Frank Wolfe. The main advantage is that the computational complexity depends

---

[*]Both authors contribute equally.

only on the sparsity of the solution, rather than the ambient dimension, i.e. it is *dimension free*. This is achieved by conducting *partial updates* in each iteration, i.e., sparse updates for $\ell_1$ and low-rank updates for the trace norm ball. While the benefits of *partial updates* is unclear for the original problem, we show in this work how they significantly benefit a primal-dual reformulation. This reduces the per iteration cost to roughly a ratio of $\frac{s}{d}$ compared to naive Frank-Wolfe, where $s$ is the sparsity (or rank) of the optimal solution, and $d$ is the feature dimension. Meanwhile, the per iteration progress of our proposal is comparable to a full gradient descent step, thus retaining linear convergence rate.

For strongly convex and smooth $f$ and $g$ we show that our algorithm achieves linear convergence with per-iteration cost $sn$ over $\ell_1$-norm ball, where $s$ upper bounds the sparsity of the primal optimal. Specifically, for sparse ERM with smooth hinge loss or quadratic loss with $\ell_2$ regularizer, our algorithm yields an overall $\mathcal{O}(s(n + \kappa) \log \frac{1}{\epsilon})$ time complexity to reach $\epsilon$ duality gap, where $\kappa$ is the condition number (smoothness divided by strong convexity). Our theory has minimal requirements on the data matrix $A$.

Experimentally we observe our method yields significantly better performance compared to prior work, especially when the data dimension is large and the solution is sparse. Therefore we achieve the state-of-the-art performance both in time complexity and in practice measured by CPU time, for regularized ERM with smooth hinge loss and matrix sensing problems.

## 2   Related Work

We review relevant algorithms that improve the overall performance of Frank-Wolfe type methods. Such improvements are roughly obtained for two reasons: the enhancement on convergence speed and the reduction on iteration cost. Very few prior works benefit in both.

Nesterov's acceleration has proven effective as in Stochastic Condition Gradient Sliding (SCGS) [23] and other variants [36, 26, 10]. Restarting techniques dynamically adapt to the function geometric properties and fills in the gap between sublinear and linear convergence for FW method [18]. Some variance reduced algorithms obtain linear convergence as in [13], however, the number of inner loops grows significantly and hence the method is not computationally efficient.

Linear convergence has been obtained specifically for polytope constraints like [27, 20], as well as the work proposed in [21, 11] that use the Away-step Frank Wolfe and Pair-wise Frank Wolfe, and their stochastic variants. One recent work [1] focuses on trace norm constraints and proposes a FW-type algorithm that yields similar progress as projected gradient descent per iteration but is almost projection free. However, in many applications where gradient computation dominates the iteration complexity, the reduction on projection step doesn't necessarily produce asymptotically better iteration costs.

The sparse update introduced by FW steps was also appreciated by [22], where they conducted dual updates with a focus on SVM with polytope constraint. Their algorithm yields low iteration costs but still suffer from sub-linear convergence.

On the other hand, the recently popularized primal-dual formulation $\min_{\boldsymbol{x}} \max_{\boldsymbol{y}} \{g(\boldsymbol{x}) + \boldsymbol{y}^\top A \boldsymbol{x} - f(\boldsymbol{y})\}$ has proven useful for different machine learning tasks like reinforcement learning, ERM, and robust optimization [8]. Especially for the ERM related problems, the primal-dual formulation still inherits the finite-sum structure from the primal form, and could be used to reduce variance [38, 35] or reduces communication complexity in the distributed setting [37]. One issue lies in this line of prior work: they do not achieve any better performance than that with the primal formulation. A notable exception is [24] where they also attempt to exploit sparsity of the primal variables with the primal-dual formulation. However, this work is for unconstrained problem so it's not directly comparable to ours. On the other hand, the analysis of [24] relies on the sparsity of the whole iterate trajectory, which has no obvious guarantee to be small. While our analysis only depends on primal optimal's sparsity or rank, and is guaranteed by the $\ell_1$ or nuclear norm constraints.

## 3   Setup

**Notation.** We briefly introduce the notation used throughout the paper. We use bold lower case letter to denote vectors, capital letter to represent matrices. $\|\cdot\|$ is $\ell_2$ norm for vectors and Frobenius norm for matrices unless specified otherwise. $\|\cdot\|_*$ indicates the trace norm for a matrix.

We say a function $f$ is $\alpha$ strongly convex if $f(\boldsymbol{y}) \geq f(\boldsymbol{x}) + \langle \boldsymbol{g}, \boldsymbol{y} - \boldsymbol{x} \rangle + \frac{\alpha}{2}\|\boldsymbol{y} - \boldsymbol{x}\|^2$, where $\boldsymbol{g} \in \partial f(\boldsymbol{x})$ is any sub-gradient of $f$. Similarly, $f$ is $\beta$-smooth when $f(\boldsymbol{y}) \leq f(\boldsymbol{x}) + \langle \boldsymbol{g}, \boldsymbol{y} - \boldsymbol{x} \rangle + \frac{\beta}{2}\|\boldsymbol{y} - \boldsymbol{x}\|^2$. We use $f^*$ to denote the convex conjugate of $f$, i.e., $f^*(\boldsymbol{y}) \overset{\text{def}}{=} \max_{\boldsymbol{x}} \langle \boldsymbol{x}, \boldsymbol{y} \rangle - f(\boldsymbol{x})$. Some more parameters are problem-specific and are defined when needed.

## 3.1 A Theoretical Vignette

To elaborate the techniques we use to obtain the linear convergence for our Frank-Wolfe type algorithm, we consider the $\ell_1$ norm constrained problem as an illustrating example:

$$\underset{\boldsymbol{x} \in \mathbb{R}^d, \|\boldsymbol{x}\|_1 \leq \tau}{\arg\min} \quad f(\boldsymbol{x}), \tag{1}$$

where $f$ is $L$-smooth and $\mu$-strongly convex. If we invoke the Frank Wolfe algorithm, we compute

$$\boldsymbol{x}^{(t)} \leftarrow (1 - \eta)\boldsymbol{x}^{(t-1)} + \eta \tilde{\boldsymbol{x}}, \quad \text{where } \tilde{\boldsymbol{x}} \leftarrow \underset{\|\boldsymbol{x}\|_1 \leq \tau}{\arg\min} \langle \nabla f(\boldsymbol{x}^{(t-1)}), \boldsymbol{x} \rangle. \tag{2}$$

Even when the function $f$ is smooth and strongly convex, (2) converges sublinearly. As inspired by [1], if we assume the optimal solution is $s$-sparse, we can enforce a sparse update while maintaining linear convergence by a mild modification on (2):

$$\boldsymbol{x}^{(t)} \leftarrow (1-\eta)\boldsymbol{x}^{(t-1)} + \eta \tilde{\boldsymbol{x}}, \text{ where } \tilde{\boldsymbol{x}} \leftarrow \underset{\|\boldsymbol{x}\|_1 \leq \tau, \|\boldsymbol{x}\|_0 \leq s}{\arg\min} \{\langle \nabla f(\boldsymbol{x}^{(t-1)}), \boldsymbol{x} \rangle + \frac{L}{2}\eta\|\boldsymbol{x}^{(t-1)} - \boldsymbol{x}\|_2^2\}. \tag{3}$$

We also call this new practice block Frank-Wolfe as in [1]. The proof of convergence can be completed within three lines. Let $h_t = f(\boldsymbol{x}^{(t)}) - f^*$.

$$h_t = f(\boldsymbol{x}^{(t-1)} + \eta(\tilde{\boldsymbol{x}} - \boldsymbol{x}^{(t-1)})) - f^*$$

$$\leq h_{t-1} + \eta \langle \nabla f(\boldsymbol{x}^{(t-1)}), \tilde{\boldsymbol{x}} - \boldsymbol{x}^{(t-1)} \rangle + \frac{L}{2}\eta^2\|\tilde{\boldsymbol{x}} - \boldsymbol{x}^{(t-1)}\|^2 \quad \text{(Smoothness of } f)$$

$$\leq h_{t-1} + \eta \langle \nabla f(\boldsymbol{x}^{(t-1)}), \boldsymbol{x}^* - \boldsymbol{x}^{(t-1)} \rangle + \frac{L}{2}\eta^2\|\boldsymbol{x}^* - \boldsymbol{x}^{(t-1)}\|^2 \quad \text{(Definition of } \tilde{\boldsymbol{x}})$$

$$\leq (1 - \eta + \frac{L}{\mu}\eta^2)h_{t-1} \quad \text{(by convexity and } \mu\text{-strong convexity of } f) \tag{4}$$

Therefore, when $\eta = \frac{\mu}{2L}$, $h_{t+1} \leq (1 - \frac{\mu}{4L})^t h_1$ and the iteration complexity is $\mathcal{O}(\frac{L}{\mu}\log(1/\epsilon))$ to achieve $\epsilon$ error.

Although we achieve linear convergence, the advantage of overall complexity against classical methods (e.g. Projected Gradient Descend (PGD)) is not shown yet. Luckily, with the update $\tilde{\boldsymbol{x}}$ being sparse, it is possible to improve the iteration complexity, while maintaining the linear convergence rate. In order to differentiate, we name the sparse update nature of (3) as *partial update*.

Next we elaborate the situations when one benefits from *partial updates*. Consider a quadratic function: $f(\boldsymbol{x}) = \frac{1}{2}\boldsymbol{x}^\top A\boldsymbol{x}$, whose gradient is $A\boldsymbol{x}$ for symmetric $A$. As $\tilde{\boldsymbol{x}}$ is sparse, One can maintain the value of the gradient efficiently: $A\boldsymbol{x}^{(t)} \equiv (1 - \eta)A\boldsymbol{x}^{(t-1)} + \eta A_{I,:}\tilde{\boldsymbol{x}}$, where $I$ is the support set of $\tilde{\boldsymbol{x}}$. We therefore reduce the complexity of one iteration to $\mathcal{O}(sd)$, compared to $\mathcal{O}(d^2)$ with PGD. Similar benefits hold when we replace $\boldsymbol{x}$ by a matrix $X$ and conduct a low-rank update on $X$. The benefit of *partial update* is not limited to quadratic functions. Next we show that for a class of composite function, we are able to take the full advantage of the *partial update*, by taking a primal-dual re-formulation.

## 4 Methodology

**Primal-Dual Formulation.** Note that the problem we are tackling is as follows:

$$\min_{\boldsymbol{x} \in C} \left\{ P(\boldsymbol{x}) \equiv \frac{1}{n}\sum_{i=1}^{n} f_i(\boldsymbol{a}_i^\top \boldsymbol{x}) + g(\boldsymbol{x}) \right\}, \tag{5}$$

We first focus on the setting where $\boldsymbol{x} \in \mathbb{R}^d$ is a vector and $C$ is the $\ell_1$-norm ball. This form covers general classification or regression tasks with $f_i$ being some loss function and $g$ being a regularizer. Extension to matrix optimization over a trace norm ball is introduced in Section 4.3.

Even with the constraint, we could reform (5) as a primal-dual convex-concave saddle point problem:

$$(5) \quad \Leftrightarrow \quad \min_{\boldsymbol{x} \in C} \max_{\boldsymbol{y}} \left\{ \mathcal{L}(\boldsymbol{x}, \boldsymbol{y}) \equiv g(\boldsymbol{x}) + \frac{1}{n}\langle \boldsymbol{y}, A\boldsymbol{x} \rangle - \frac{1}{n}\sum_{i=1}^{n} f_i^*(y_i) \right\}, \tag{6}$$

or its dual formulation:

$$(5) \Leftrightarrow \max_{\boldsymbol{y}} \left\{ D(\boldsymbol{y}) := \min_{\boldsymbol{x} \in C} \left\{ g(\boldsymbol{x}) + \frac{1}{n} \langle \boldsymbol{y}, A\boldsymbol{x} \rangle \right\} - \frac{1}{n} \sum_{i=1}^{n} f_i^*(y_i) \right\}. \tag{7}$$

Notice (7) is not guaranteed to have an explicit form. Therefore some existing FW variants like [22] that optimizes over (7) may not apply. Instead, we directly solve the convex concave problem (6) and could therefore solve more general problems, including complicated constraint like trace norm.

Since the computational cost of the gradient $\nabla_{\boldsymbol{x}} \mathcal{L}$ and $\nabla_{\boldsymbol{y}} \mathcal{L}$ is dominated by computing $A^\top \boldsymbol{y}$ and $A\boldsymbol{x}$ respectively, *sparse updates* could reduce computational costs by a ratio of roughly $\mathcal{O}(d/s)$ for updating $\boldsymbol{x}$ and $\boldsymbol{y}$ while achieving good progress.

## 4.1 Primal-Dual Block Generalized Frank-Wolfe

With the primal-dual formulation, we are ready to introduce our algorithm. The idea is simple: since the primal variable $\boldsymbol{x}$ is constrained over $\ell_1$ norm ball, we conduct block Frank-Wolfe algorithm and achieve an $s$-sparse update. Meanwhile, for the dual variable $\boldsymbol{y}$ we conduct greedy coordinate ascent method to select and update $k$ coordinates ($k$ determined later). We selected coordinates that allow the largest step, which is usually referred as a Gauss-Southwell rule denoted by **GS-r** [30]. Our algorithm is formally presented in Algorithm 1. We have the following assumptions on $f$ and $g$:

**Assumption 4.1.** *We assume the functions satisfy the following properties:*

- *Each loss function $f_i$ is convex and $\beta$-smooth, and is $\alpha$ strongly convex over some convex set (could be $\mathbb{R}$), and linear otherwise.*
- $\max_i \|\boldsymbol{a}_i\|_2^2 \leq R$. *Therefore $\frac{1}{n} \sum_{i=1}^{n} f_i(\boldsymbol{a}_i^\top \boldsymbol{x})$ is $\beta R$-smooth.*
- *$g$ is $\mu$-strongly convex and $L$-smooth.*

Suitable loss functions $f_i$ include smooth hinge loss [33] and quadratic loss function. Relevant applications covered are Support Vector Machine (SVM) with smooth hinge loss, elastic net [39], and linear regression problem with quadratic loss.

---

**Algorithm 1** Primal-Dual Block Generalized Frank-Wolfe Method for $\ell_1$ Norm Ball

---

1: **Input:** Training data $A \in \mathbb{R}^{n \times d}$, primal and dual step size $\eta, \delta > 0$.
2: **Initialize:** $\boldsymbol{x}^{(0)} \leftarrow 0 \in \mathbb{R}^d$, $\boldsymbol{y}^{(0)} \leftarrow 0 \in \mathbb{R}^n$, $\boldsymbol{w}^{(0)} \equiv A\boldsymbol{x} = 0 \in \mathbb{R}^n$, $\boldsymbol{z}^{(0)} \equiv A^\top \boldsymbol{y} = 0 \in \mathbb{R}^d$
3: **for** $t = 1, 2, \cdots, T$ **do**
4:     Use Block Frank Wolfe to update the primal variable:

$$\tilde{\boldsymbol{x}} \leftarrow \operatorname*{arg\,min}_{\|\boldsymbol{x}\|_1 \leq \lambda, \|\boldsymbol{x}\|_0 \leq s} \left\{ \langle \frac{1}{n} \boldsymbol{z}^{(t-1)} + \nabla g(\boldsymbol{x}^{(t-1)}), \boldsymbol{x} \rangle + \frac{L}{2} \eta \|\boldsymbol{x} - \boldsymbol{x}^{(t-1)}\|^2 \right\} \tag{8}$$

$$\boldsymbol{x}^{(t)} \leftarrow (1 - \eta)\boldsymbol{x}^{(t-1)} + \eta\tilde{\boldsymbol{x}}$$

5:     Update $w$ to maintain the value of $A\boldsymbol{x}$:

$$\boldsymbol{w}^{(t)} \leftarrow (1 - \eta)\boldsymbol{w}^{(t-1)} + \eta A \Delta \boldsymbol{x} \tag{9}$$

6:     Consider the potential dual update:

$$\tilde{\boldsymbol{y}} = \operatorname*{arg\,max}_{\boldsymbol{y}'} \left\{ \frac{1}{n} \langle \boldsymbol{w}^{(t)}, \boldsymbol{y}' \rangle - f^*(\boldsymbol{y}') - \frac{1}{2\delta} \|\boldsymbol{y}' - \boldsymbol{y}^{(t-1)}\|^2 \right\}. \tag{10}$$

7:     Choose greedily the dual coordinates to update: let $I^{(t)}$ be the top $k$ coordinates that maximize

$$|\tilde{y}_i - y_i^{(t-1)}|, i \in [n].$$

    Update the dual variable accordingly:

$$y_i^{(t)} \leftarrow \begin{cases} \tilde{y}_i & \text{if } i \in I^{(t)} \\ y_i^{(t-1)} & \text{otherwise.} \end{cases} \tag{11}$$

8:     Update $z$ to maintain the value of $A^\top \boldsymbol{y}$

$$\boldsymbol{z}^{(t)} \leftarrow \boldsymbol{z}^{(t-1)} + A_{:,I^{(t)}}^\top (\boldsymbol{y}^{(t)} - \boldsymbol{y}^{(t-1)}) \tag{12}$$

9: **end for**
10: **Output:** $\boldsymbol{x}^{(T)}, \boldsymbol{y}^{(T)}$

---

As $\mathcal{L}(\boldsymbol{x}, \boldsymbol{y})$ is $\mu$-strongly convex and $L$-smooth with respect to $\boldsymbol{x}$, we set the primal learning rate $\eta = \frac{\mu}{2L}$ according to Section 3.1. Meanwhile, the dual learning rate $\delta$ is set to balance its effect on the dual progress as well as the primal progress. We specify it in the theoretical analysis part.

The computational complexity for each iteration in Algorithm 1 is $\mathcal{O}(ns)$. Both primal and dual update could be viewed as roughly three steps: coordinate selection, variable update, and maintaining $A^T\boldsymbol{y}$ or $A\boldsymbol{x}$. The coordinate selection as Eqn. (8) for primal and the choice of $I^{(t)}$ for dual variable respectively take $\mathcal{O}(d)$ and $\mathcal{O}(n)$ on average if implemented with the quick selection algorithm. The variable update costs $\mathcal{O}(d)$ and $\mathcal{O}(n)$. The dominating cost is to maintain $A\boldsymbol{x}$ as in Eqn. (9) that takes $\mathcal{O}(ns)$, and $\mathcal{O}(dk)$ of maintaining $A^\top\boldsymbol{y}$ as in Eqn. (12). To balance the time budget for primal and dual step, we set $k = ns/d$ and achieve an overall complexity of $\mathcal{O}(ns)$ per iteration.

## 4.2 Theoretical Analysis

We derive convergence analysis under Assumption 4.1. The derivation consists of the analysis on the primal progress, the balance of the dual progress, and their overall effect.

Define the primal gap as $\Delta_p^{(t)} \stackrel{\text{def}}{=} \mathcal{L}(\boldsymbol{x}^{(t+1)}, \boldsymbol{y}^{(t)}) - \mathcal{L}(\bar{\boldsymbol{x}}^{(t)}, \boldsymbol{y}^{(t)})$, where $\bar{\boldsymbol{x}}^{(t)}$ is the primal optimal solution such that the dual $D(\boldsymbol{y}^{(t)}) = \mathcal{L}(\bar{\boldsymbol{x}}^{(t)}, \boldsymbol{y}^{(t)})$, and is sparse enforced by the $\ell_1$ constraint. The dual gap is $\Delta_d^{(t)} \stackrel{\text{def}}{=} D^* - D(\boldsymbol{y}^{(t)})$. We analyze the convergence rate of duality gap $\Delta^{(t)} \equiv \max\{1, (\beta/\alpha - 1)\}\Delta_P^{(t)} + \Delta_d^{(t)}$.

**Primal progress:** Firstly, similar to the analysis in Section 3.1, we could derive that primal update introduces a sufficient descent as in Lemma A.2.
$$\mathcal{L}(\boldsymbol{x}^{(t+1)}, \boldsymbol{y}^{(t)}) - \mathcal{L}(\boldsymbol{x}^{(t)}, \boldsymbol{y}^{(t)}) \le -\frac{\eta}{2}\Delta_p^{(t)}.$$

**Dual progress:** With the **GS-r** rule to carefully select and update the most important $k$ coordinates in the dual variable in (10), we are able to derive the following result on dual progress that diminishes dual gap as well as inducing error.
$$-\|\boldsymbol{y}^{(t)} - \boldsymbol{y}^{(t-1)}\|^2 \le -\frac{k\delta}{n\beta}\Delta_d^{(t)} + \frac{k\delta}{n^2}R\|\bar{\boldsymbol{x}}^{(t)} - \boldsymbol{x}^{(t)}\|_2^2$$
Refer to Lemma A.5 for details.

**Primal Dual progress:** The overall progress evolves as:
$$\Delta^{(t)} - \Delta^{(t-1)} \le \overbrace{\mathcal{L}(\boldsymbol{x}^{(t+1)}, \boldsymbol{y}^{(t)}) - \mathcal{L}(\boldsymbol{x}^{(t)}, \boldsymbol{y}^{(t)})}^{\text{primal progress}} - \frac{1}{4\delta}\overbrace{\|\boldsymbol{y}^{(t)} - \boldsymbol{y}^{(t-1)}\|^2}^{\text{dual progress}} + \frac{3\delta Rk}{2n^2}\overbrace{\|\bar{\boldsymbol{x}}^{(t)} - \boldsymbol{x}^{(t)}\|^2}^{\text{primal hindrance}}.$$
In this way, we are able to connect the progress on duality gap with constant fraction of its value, and achieve linear convergence:

**Theorem 4.1.** *Given a function $P(\boldsymbol{x}) = \sum_{i=1}^n f_i(\boldsymbol{a}_i^\top \boldsymbol{x}) + g(\boldsymbol{x})$ that satisfies Assumption 4.1. Set $s$ to upper bound the sparsity of the primal optimal $\bar{\boldsymbol{x}}^{(t)}$, and learning rates $\eta = \frac{\mu}{2L}, \delta = \frac{1}{k}(\frac{L}{\mu n\beta} + \frac{5\beta R}{2\alpha\mu n^2}(1 + 4\frac{L}{\mu}))^{-1}$. The duality gap $\Delta^{(t)} = \max\{1, \frac{\beta}{\alpha} - 1\}\Delta_P^{(t)} + \Delta_d^{(t)}$ generated by Algorithm 1 takes $\mathcal{O}(\frac{L}{\mu}(1 + \frac{\beta}{\alpha}\frac{R\beta}{n\mu})\log\frac{1}{\epsilon})$ iterations to achieve $\epsilon$ error. The overall complexity is $\mathcal{O}(ns\frac{L}{\mu}(1 + \frac{\beta}{\alpha}\frac{R\beta}{n\mu})\log\frac{1}{\epsilon})$.*

For our target applications like elastic net, or ERM with smooth hinge loss, we are able to connect the time complexity to the condition number of the primal form.

**Corollary 4.1.1.** *Given a smooth hinge loss or quadratic loss $f_i$ that is $\beta$-smooth, and $\ell_2$ regularizer $g = \frac{\mu}{2}\|\boldsymbol{x}\|^2$. Define the condition number $\kappa = \frac{\beta R}{\mu}$. Setting $s$ upper bounds the sparsity of the primal optimal $\bar{\boldsymbol{x}}^{(t)}$, and learning rates $\eta = \frac{1}{2}, \delta = \frac{1}{k}(\frac{1}{n\beta} + \frac{25R}{2\mu n^2})^{-1}$, the duality gap $\Delta^{(t)}$ takes $\mathcal{O}((1 + \frac{\kappa}{n})\log\frac{1}{\epsilon})$ iterations to achieve $\epsilon$ error. The overall complexity is $\mathcal{O}(s(n + \kappa)\log\frac{1}{\epsilon})$.*

Our derivation of overall complexity implicitly requires $ns \ge d$ by setting $k = sd/n \ge 1$. This is true for our considered applications like SVM. Otherwise we choose $k = 1$ and the complexity becomes $\mathcal{O}(\max\{d, ns\}(1 + \frac{\kappa}{n})\log\frac{1}{\epsilon})$.

In Table 1, we briefly compare the time complexity of our algorithm with some benchmark algorithms: (1) Accelerated Projected Gradient Descent (PGD) (2) Frank-Wolfe algorithm (FW) (3) Stochastic Variance Reduced Gradient (SVRG) [15] (4) Stochastic Conditional Gradient Sliding (SCGS) [23] and (5) Stochastic Variance-Reduced Conditional Gradient Sliding (STORC) [13]. The comparison is not thorough but intends to select constrained optimization that improves the overall complexity from different perspective. Among them, accelerated PGD improves conditioning of the problem,

while SCGS and STORC reduces the dependence on number of samples. In the experimental session we show that our proposal outperforms the listed algorithms under various conditions.

| Algorithm | Per Iteration Cost | Iteration Complexity |
|---|---|---|
| Frank Wolfe | $\mathcal{O}(nd)$ | $\mathcal{O}(\frac{1}{\epsilon})$ |
| Accelerated PGD [29] | $\mathcal{O}(nd)$ | $\mathcal{O}(\sqrt{\kappa}\log\frac{1}{\epsilon})$ |
| SVRG [15] | $\mathcal{O}(nd)$ | $\mathcal{O}((1+\kappa/n)\log\frac{1}{\epsilon})$ |
| SCGS [23] | $\mathcal{O}(\kappa^2\frac{\#\text{iter}^3}{\epsilon^2}d)$ | $\mathcal{O}(\frac{1}{\epsilon})$ |
| STORC [13] | $\mathcal{O}(\kappa^2 d + nd)$ | $\mathcal{O}(\log\frac{1}{\epsilon})$ |
| Primal Dual FW (ours) | $\mathcal{O}(ns)$ | $\mathcal{O}((1+\kappa/n)\log\frac{1}{\epsilon})$ |

Table 1: Time complexity comparisons on the setting of Corollary 4.1.1. For clear comparison, we refer the per iteration cost as the time complexity of outer iterations.

## 4.3 Extension to the Trace Norm Ball

---
**Algorithm 2** Primal-Dual Block Generalized Frank-Wolfe Method for Trace Norm Ball
---
1: **Input:** Training data $A \in \mathbb{R}^{n\times d}$, primal and dual step size $\eta, \delta > 0$. Target accuracy $\epsilon$.
2: **Initialize:** $X^{(0)} \leftarrow 0 \in \mathbb{R}^{d\times c}, Y^{(0)} \leftarrow 0 \in \mathbb{R}^{n\times c}, W^{(0)} \equiv AX = 0 \in \mathbb{R}^{n\times c}, Z^{(0)} \equiv A^\top Y = 0 \in \mathbb{R}^{d\times c}$
3: **for** $t = 1, 2, \cdots, T$ **do**
4:   Use Frank Wolfe to Update the primal variable:
$$X^{(t)} \leftarrow (1-\eta)X^{(t-1)} + \eta\tilde{X}, \text{ where } \tilde{X} \leftarrow (\frac{1}{2}, \frac{\epsilon}{8})\text{-approximation of Eqn. (18)}.$$

5:   Update $W$ to maintain the value of $AX$:
$$W^{(t)} \leftarrow (1-\eta)W^{(t-1)} + \eta A\tilde{X} \qquad (13)$$

6:   Consider the potential dual update:
$$\tilde{Y}^{(t)} \leftarrow \arg\max_Y \left\{ \langle W, Y\rangle - f^*(Y) - \frac{1}{2\delta}\|Y - Y^{(t-1)}\|^2 \right\} \qquad (14)$$

7:   Choose greedily the rows of the dual variable to update: let $I^{(t)}$ be the top $k$ coordinates that maximize
$$\left\|\tilde{Y}_{i,:} - Y_{i,:}^{(t-1)}\right\|_2, i \in [n].$$
  Update the dual variable accordingly:
$$Y_{i,:}^{(t)} \leftarrow \begin{cases} \tilde{Y}_{i,:} & \text{if } i \in I^{(t)} \\ Y_{i,:}^{(t-1)} & \text{otherwise.} \end{cases} \qquad (15)$$

8:   Update $Z$ to maintain the value of $A^\top Y$
$$Z^{(t)} \leftarrow Z^{(t-1)} + A^\top(Y^{(t)} - Y^{(t-1)}) \qquad (16)$$

9: **end for**
10: **Output:** $X^{(T)}, Y^{(T)}$
---

We also extend our algorithm to matrix optimization over trace norm constraints:
$$\min_{\|X\|_*\leq\lambda, X\in\mathbb{R}^{d\times c}} \left\{ \frac{1}{n}\sum_{i=1}^n f_i(\boldsymbol{a}_i^\top X) + g(X) \right\}. \qquad (17)$$
This formulation covers multi-label multi-class problems, matrix completion, affine rank minimization, and phase retrieval problems (see reference therein [3, 1]). Equivalently, we solve the following primal-dual problem:
$$\min_{\|X\|_*\leq\lambda, X\in\mathbb{R}^{d\times c}} \max_{Y\in\mathbb{R}^{n\times c}} \left\{ \mathcal{L}(X,Y) \equiv g(X) + \frac{1}{n}\langle AX, Y\rangle - \frac{1}{n}\sum_{i=1}^n f_i^*(\boldsymbol{y}_i) \right\}.$$
Here $\boldsymbol{y}_i$ is the $i$-th row of the dual matrix $Y$. For this problem, the *partial update* we enforced on the primal matrix is to keep the update matrix low rank:
$$\tilde{X} \leftarrow \arg\min_{\|X\|_*\leq\lambda, \text{rank}(X)\leq s} \left\{ \langle \frac{1}{n}Z + \nabla g(X^{(t-1)}), X\rangle + \frac{L}{2}\eta\|X - X^{(t-1)}\|^2 \right\}, Z \equiv A^\top Y^{(t-1)}. \quad (18)$$

However, an exact solution to (18) requires computing the top $s$ left and right singular vectors of the matrix $X^{(t-1)} - \frac{1}{\eta L}(Z + \nabla g(X^{(t-1)})) \in \mathbb{R}^{d \times c}$. Therefore we loosely compute an $(\frac{1}{2}, \epsilon/2)$-approximation, where $\epsilon$ is the target accuracy, based on the following definition:

**Definition 4.2** (Restated Definition 3.2 in [1]). *Let $l_t(V) = \langle \nabla_X \mathcal{L}(X^{(t)}, Y^{(t)}), V - X^{(t)} \rangle + \frac{L}{2}\eta \|V - X^{(t)}\|_F^2$ be the objective function in (18), and let $l_t^* = l_t(\bar{X}^{(t)})$. Given parameters $\gamma \geq 0$ and $\epsilon \geq 0$, a feasible solution $V$ to (18) is called $(\gamma, \epsilon)$-approximate if it satisfies $l(V) \leq (1 - \gamma)l_t^* + \epsilon$.*

The time dependence on the data size $n, c, d, s$ is $ncs + s^2(n + c)$ [1], and is again independent of $d$. Meanwhile, the procedures to keep track of $W^{(t)} \equiv AX^{(t)}$ requires complexity of $nds + ncs$, while updating $Y^{(t)}$ requires $dck$ operations. Therefore, by setting $k \leq ns(1/c + 1/d)$, the iteration complexity's dependence on the data size becomes $\mathcal{O}(n(d + c)s)$ operations, instead of $\mathcal{O}(ndc)$ for conducting a full projected gradient step. Recall that $s$ upper bounds the rank of $\bar{X}^{(t)} \leq \min\{d, c\}$.

The trace norm version mostly inherits the convergence guarantees for vector optimization. Refer to the Appendix for details.

**Assumption 4.2.** *We assume the following property on the primal form* (17)*:*

- *$f_i$ is convex, and $\beta$-smooth. Its convex conjugate $f_i^*$ exists and satisfies $\frac{1}{\alpha}$-smooth on some convex set (could be $\mathbb{R}^c$) and infinity otherwise.*
- *Data matrix $A$ satisfies $R = \max_{|I| \leq k, I \subset [n]} \sigma_{\max}^2(A_{I,:})$ ($\leq \|A\|_2^2$). Here $\sigma_{\max}(X)$ denotes the largest singular value of $X$.*
- *$g$ is $\mu$-strongly convex and $L$-smooth.*

The assumptions also cover smooth hinge loss as well as quadratic loss. With the similar assumptions, the convergence analysis for Algorithm 2 is almost the same as Algorithm 1. The only difference comes from the primal step where approximated update produces some error:

**Primal progress:** With the primal update rule in Algorithm 2, it satisfies $\mathcal{L}(X^{(t+1)}, Y^{(t)}) - \mathcal{L}(X^{(t)}, Y^{(t)}) \leq -\frac{\mu}{8L}\Delta_p^{(t)} + \frac{\epsilon}{16}$. (See Lemma A.7.) With no much modification in the proof, we are able to derive similar convergence guarantees for the trace norm ball.

**Theorem 4.3.** *Given a function $\frac{1}{n}\sum_{i=1}^{n} f_i(\boldsymbol{a}_i^\top X) + g(X)$ that satisfies Assumption 4.2. Setting $s \geq rank(\bar{X}^{(t)})$, and learning rate $\eta = \frac{\mu}{2L}, \delta \leq \frac{1}{k}(\frac{L}{\mu n \beta} + \frac{5\beta R}{2\alpha \mu n^2}(1 + 8\frac{L}{\mu}))^{-1}$, the duality gap $\Delta^{(t)}$ generated by Algorithm 2 satisfies $\Delta^{(t)} \leq \frac{k\delta}{k\delta + 8\beta n}\Delta^{(t-1)} + \frac{\epsilon}{16}$. Therefore it takes $\mathcal{O}(\frac{L}{\alpha}(1 + \frac{\beta}{\alpha}\frac{R\beta}{n\mu})\log\frac{1}{\epsilon})$ iterations to achieve $\epsilon$ error.*

We also provide a brief analysis on the difficulty to extend our algorithm to polytope-type constraints in the Appendix A.9.

# 5 Experiments

We evaluate the Primal-Dual Block Generalized Frank-Wolfe algorithm by its performance on binary classification with smoothed hinge loss[2]. We refer the readers to Appendix A.7 for details about smoothed hinge loss.

We compare the proposed algorithm against five benchmark algorithms: (1) Accelerated Projected Gradient Descent (Acc PG) (2) Frank-Wolfe algorithm (FW) (3) Stochastic Variance Reduced Gradient (SVRG) [15] (4) Stochastic Conditional Gradient Sliding (SCGS) [23] and (5) Stochastic Variance-Reduced Conditional Gradient Sliding (STORC) [13]. We presented the time complexity for each algorithm in Table 1. Three of them (FW, SCGS, STORC) are projection-free algorithms, and the other two (Acc PG, SVRG) are projection-based algorithms. Algorithms are implemented in C++, with the Eigen linear algebra library [12].

The six datasets used here are summarized in Table 2. All of them can be found in LIBSVM datasets [4]. We augment the features of MNIST, ijcnn, and cob-rna by random binning [32], which is a standard technique for kernel approximation. Data is normalized. We set the $\ell_1$ constraint to be 300 and the $\ell_2$ regularize parameter to $10/n$ to achieve reasonable prediction accuracy. We refer the

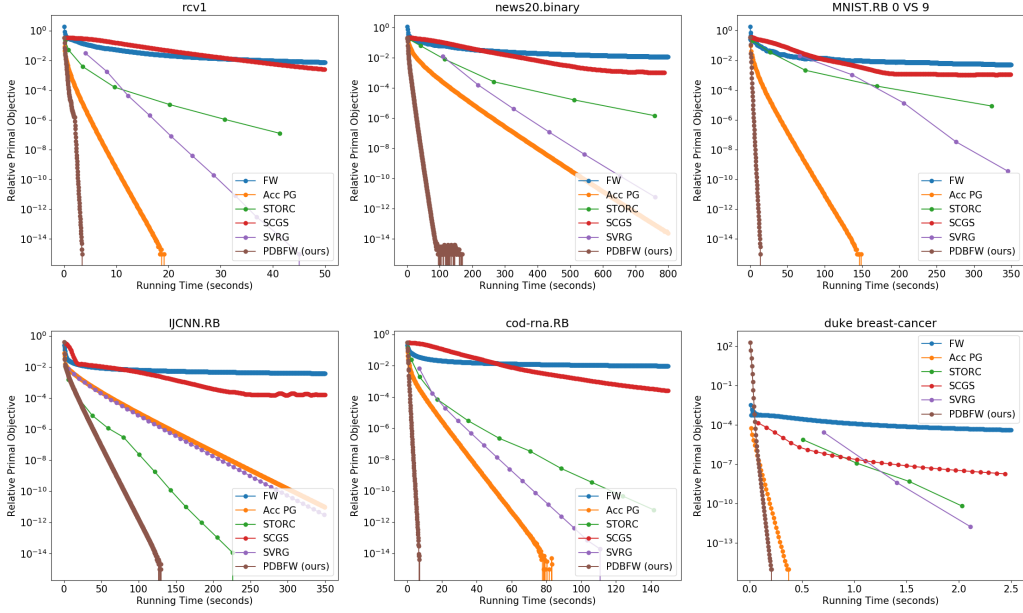

Figure 1: **Convergence result comparison of different algorithms on smoothed hinge loss.** For six different datasets, we show the decrease of relative primal objective: $(P(\boldsymbol{x}^{(t)}) - P^*)/P^*$ over CPU time. Our algorithm (brown) achieves around 10 times speedup over all other methods except for the smallest dataset duke.

| Dataset Name | # Features | # Samples | # Non-Zero | Solution Sparsity (Ratio) |
|---|---|---|---|---|
| duke breast-cancer [4] | 7,129 | 44 | 313,676 | 423 (5.9%) |
| rcv1 [4] | 47,236 | 20,242 | 1,498,952 | 1,169 (2.5%) |
| news20.binary [4] | 1,355,191 | 19,996 | 9,097,916 | 1,365 (0.1%) |
| MNIST.RB 0 VS 9 [4, 32] | 894,499 | 11,872 | 1,187,200 | 8,450 (0.9%) |
| ijcnn.RB [4, 32] | 58,699 | 49,990 | 14,997,000 | 715 (1.2%) |
| cob-rna.RB [4, 32] | 81,398 | 59,535 | 5,953,500 | 958 (1.2%) |

Table 2: Summary of the properties of the datasets.

readers to the Appendix C.1 for results of other choice of parameters. These datasets have various scale of features, samples, and solution sparsity ratio.

The results are shown in Fig 1. To focus on the convergence property, we show the decrease of loss function instead of prediction accuracy. From Fig 1, our proposed algorithm consistently outperforms the benchmark algorithms. The winning margin is roughly proportional to the solution sparsity ratio, which is consistent with our theory.

We also implement Algorithm 2 for trace norm ball and compare it with some prior work in the Appendix C.2, especially Block FW [1]. We generated synthetic data with optimal solutions of different ranks, and show that our proposal is consistently faster than others.

## 6 Conclusion

In this paper we consider a class of problems whose solutions enjoy some simple structure induced by the constraints. We propose a FW type algorithm to exploit the simple structure and conduct partial updates, reducing the time cost for each update remarkably while attaining linear convergence. For a class of ERM problems, our running time depends on the sparsity/rank of the optimal solutions rather than the ambient feature dimension. Our empirical studies verify the improved performance compared to various state-of-the-art algorithms.

**Acknowledgements.** This work is supported by NSF Grants 1618689, EECS-1609279, CCF-1302435, CNS-1704778, IIS-1546452, CCF-1564000, DMS 1723052, CCF 1763702, AF 1901292 and research gifts by Google, Western Digital and NVIDIA.

## Footnotes

[2]The codes to reproduce our results could be found in `https://github.com/CarlsonZhuo/primal_dual_frank_wolfe`.

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
