[Supplementary Material]

# A Omitted Proofs

## A.1 Derivation of Primal-Dual Formulation

$$\min_{\boldsymbol{x}\in C} P(\boldsymbol{x}) = \frac{1}{n}\sum_{i=1}^{n} f_i(\boldsymbol{a}_i^\top \boldsymbol{x}) + g(\boldsymbol{x})$$

$$= \min_{\boldsymbol{x}\in C, \boldsymbol{b}=A\boldsymbol{x}} \frac{1}{n}\sum_{i=1}^{n} f_i(b_i) + g(\boldsymbol{x})$$

$$= \min_{\boldsymbol{x}\in C, \boldsymbol{b}} \max_{\boldsymbol{y}} \left\{ \frac{1}{n}\sum_{i=1}^{n} f_i(b_i) + g(\boldsymbol{x}) + \frac{1}{n}\langle \boldsymbol{y}, A\boldsymbol{x} - \boldsymbol{b}\rangle \right\}$$

$$= \min_{\boldsymbol{x}\in C} \max_{\boldsymbol{y}} \left\{ g(\boldsymbol{x}) + \frac{1}{n}\langle \boldsymbol{y}, A\boldsymbol{x}\rangle + \min_{\boldsymbol{b}} \left\{ \frac{1}{n}\sum_{i=1}^{n} f_i(b_i) - \frac{1}{n}\langle \boldsymbol{y}, \boldsymbol{b}\rangle \right\} \right\}$$

$$= \min_{\boldsymbol{x}\in C} \max_{\boldsymbol{y}} \left\{ \mathcal{L}(\boldsymbol{x}, \boldsymbol{y}) := g(\boldsymbol{x}) + \frac{1}{n}\langle \boldsymbol{y}, A\boldsymbol{x}\rangle - \frac{1}{n}\sum_{i=1}^{n} f_i^*(y_i) \right\}$$

$$= \max_{\boldsymbol{y}} \left\{ D(\boldsymbol{y}) := \min_{\boldsymbol{x}\in C} \left\{ g(\boldsymbol{x}) + \frac{1}{n}\langle \boldsymbol{y}, A\boldsymbol{x}\rangle - \frac{1}{n}\sum_{i=1}^{n} f_i^*(y_i) \right\} \right\}$$

We use Von Neumann-Fan minimax theorem for the whole derivation when swapping each min-max formula [7]. For the last equality, there is a convex constraint in the minimization part. Although the original Von Neumann-Fan doesn't have constraints, it naturally applies to the case when $\boldsymbol{x}$ (assuming function is convex to $\boldsymbol{x}$) is bounded in a convex set, since we could change $f(\boldsymbol{x}, \boldsymbol{y})$ to $f(\boldsymbol{x}, \boldsymbol{y}) + I_C(\boldsymbol{x})$, where $I_C(\boldsymbol{x}) = 0$ if $\boldsymbol{x} \in C$ and $\infty$ otherwise. Then the property will be properly inherited.

## A.2 Notation and simple facts

Recall primal, dual and Lagrangian forms:

$$P(\boldsymbol{x}) \stackrel{\text{def}}{=} \frac{1}{n}\sum_{i=1}^{n} f_i(\boldsymbol{a}_i^\top \boldsymbol{x}) + g(\boldsymbol{x})$$

$$\mathcal{L}(\boldsymbol{x}, \boldsymbol{y}) \stackrel{\text{def}}{=} g(\boldsymbol{x}) + \frac{1}{n}\boldsymbol{y}^\top A\boldsymbol{x} - \frac{1}{n}\sum_{i=1}^{n} f_i^*(y_i)$$

$$D(\boldsymbol{y}) \stackrel{\text{def}}{=} \min_{\boldsymbol{x}\in C} \mathcal{L}(\boldsymbol{x}, \boldsymbol{y}) = \mathcal{L}(\bar{\boldsymbol{x}}(\boldsymbol{y}), \boldsymbol{y})$$

Similar to the definitions in [24], we introduce the primal gap defined as $\Delta_p^{(t)} \stackrel{\text{def}}{=} \mathcal{L}(\boldsymbol{x}^{(t+1)}, \boldsymbol{y}^{(t)}) - D(\boldsymbol{y}^{(t)})$, and dual gap $\Delta_d^{(t)} \stackrel{\text{def}}{=} D^* - D(\boldsymbol{y}^{(t)})$. Recall the assumptions:

- $f_i$ is convex and $\beta$-smooth, and is $\alpha$ strongly convex over some convex set, and linear otherwise.
- $R = \max_i \|\boldsymbol{a}_i\|_2^2, \forall i \in [n]$.
- $g$ is $\mu$-strongly convex and $L$-smooth.

To begin with, it is easy to verify that $f_i^*$ is $1/\beta$-strongly convex and is $1/\alpha$-smooth on a convex set and infinity otherwise (See Claim A.6). For simplicity we first assume $\alpha \geq \frac{1}{2}\beta$ and then generalize the result.

**Claim A.1.**
- *Since $D(\boldsymbol{y}) = \min_{\boldsymbol{x}\in C}\{g(\boldsymbol{x}) + \frac{1}{n}\boldsymbol{y}^\top A\boldsymbol{x}\} - \frac{1}{n}\sum_{i=1}^{n} f_i^*(\boldsymbol{y})$, $-D(\boldsymbol{y})$ is at least $\frac{1}{\beta}$-strongly convex.*

- *Based on our update rule, $\exists \boldsymbol{g} \in \partial_{\boldsymbol{y}} \frac{1}{n}\sum_i f_i^*(\boldsymbol{y}^{(t)})$, such that*

$$\boldsymbol{y}_{I^{(t)}}^{(t)} - \boldsymbol{y}_{I^{(t)}}^{(t-1)} = \delta(\frac{1}{n}A_{I^{(t)},:}\boldsymbol{x}^{(t)} - \boldsymbol{g}_{I^{(t)}}). \tag{19}$$

*And our update rule ensures that $I^{(t)}$ consists of indices $i \in [n]$ that maximizes $|\frac{1}{n} \boldsymbol{a}_i^\top \boldsymbol{x}^{(t)} - g_i|$.*

## A.3 Primal Progress

**Lemma A.2.** *(Primal Progress)*
$$\mathcal{L}(\boldsymbol{x}^{(t+1)}, \boldsymbol{y}^{(t)}) - \mathcal{L}(\bar{\boldsymbol{x}}^{(t)}, \boldsymbol{y}^{(t)}) \leq (1 - \frac{\eta}{2}) \left( \mathcal{L}(\boldsymbol{x}^{(t)}, \boldsymbol{y}^{(t)}) - \mathcal{L}(\bar{\boldsymbol{x}}^{(t)}, \boldsymbol{y}^{(t)}) \right)$$
*Or equivalently,*
$$(1 - \frac{\eta}{2})(\mathcal{L}(\boldsymbol{x}^{(t+1)}, \boldsymbol{y}^{(t)}) - \mathcal{L}(\boldsymbol{x}^{(t)}, \boldsymbol{y}^{(t)})) \leq -\frac{\eta}{2} \left( \mathcal{L}(\boldsymbol{x}^{(t+1)}, \boldsymbol{y}^{(t)}) - \mathcal{L}(\bar{\boldsymbol{x}}^{(t)}, \boldsymbol{y}^{(t)}) \right) \equiv -\frac{\eta}{2} \Delta_p^{(t)}$$

*Proof.* Simply replace $h_t$ as $\mathcal{L}(\boldsymbol{x}^{(t)}, \boldsymbol{y}^{(t)}) - D(\boldsymbol{y}^{(t)})$ and $h_{t+1}$ as $\mathcal{L}(\boldsymbol{x}^{(t+1)}, \boldsymbol{y}^{(t)}) - D(\boldsymbol{y}^{(t)})$ in Inequality (4). We could conclude that $h_{t+1} \leq (1 - \eta + \eta^2 \frac{L}{\mu}) h_t$. Therefore when $\eta \leq \frac{\mu}{2L}$, $h_{t+1} \leq (1 - \frac{\eta}{2}) h_t$ and the first part of Lemma A.2 is true. Some simple rearrangement suffices the second part of the lemma. $\square$

## A.4 Primal Dual Progress

In order to get a clue on how to analyze the dual progress, we first look at how the primal and dual evolve through iterations.

For an index set $I$ and a vector $\boldsymbol{y} \in \mathbb{R}^n$, denote $\boldsymbol{y}_I = \sum_{i \in I} y_i \boldsymbol{e}_i \in \mathbb{R}^k$ as the subarray of $\boldsymbol{y}$ indexed by $I$, with $|I| = k$. Recall Algorithm 1 selects the coordinates to update in the dual variable as $I^{(t)}$.

**Lemma A.3.** *(Primal-Dual Progress).*
$$\Delta_d^{(t)} - \Delta_d^{(t-1)} + \Delta_p^{(t)} - \Delta_p^{(t-1)}$$
$$\leq \mathcal{L}(\boldsymbol{x}^{(t+1)}, \boldsymbol{y}^{(t)}) - \mathcal{L}(\boldsymbol{x}^{(t)}, \boldsymbol{y}^{(t)}) - \frac{1}{2\delta} \|\boldsymbol{y}^{(t)} - \boldsymbol{y}^{(t-1)}\|^2$$
$$+ \frac{2\delta R k}{n^2} \|\bar{\boldsymbol{x}}^{(t)} - \boldsymbol{x}^{(t)}\|^2.$$

*Proof.* Notice we have claimed that $-D(\boldsymbol{y})$ is $\frac{1}{\beta}$-strongly convex and for all $\boldsymbol{g} \in \partial_{\boldsymbol{y}} \frac{1}{n} \sum_i^n f_i^*(\boldsymbol{y}^{(t)})$,
$$\Delta_d^{(t)} - \Delta_d^{(t-1)} = \left( -D(\boldsymbol{y}^{(t)}) \right) - \left( -D(\boldsymbol{y}^{(t-1)}) \right)$$
$$\leq \langle -\nabla_{\boldsymbol{y}} \mathcal{L}(\bar{\boldsymbol{x}}^{(t)}, \boldsymbol{y}^{(t)}), \boldsymbol{y}^{(t)} - \boldsymbol{y}^{(t-1)} \rangle - \frac{1}{2\beta} \|\boldsymbol{y}^{(t)} - \boldsymbol{y}^{(t-1)}\|^2$$
$$= -\langle \frac{1}{n} A_{I^{(t)}, :} \bar{\boldsymbol{x}}^{(t)} - \boldsymbol{g}_{I^{(t)}}, \boldsymbol{y}_{I^{(t)}}^{(t)} - \boldsymbol{y}_{I^{(t)}}^{(t-1)} \rangle - \frac{1}{2\beta} \|\boldsymbol{y}^{(t)} - \boldsymbol{y}^{(t-1)}\|^2 \qquad (20)$$
Meanwhile since $-\mathcal{L}(\boldsymbol{x}, \boldsymbol{y})$ is $\frac{1}{\alpha}$-smooth over its feasible set,
$$\mathcal{L}(\boldsymbol{x}^{(t)}, \boldsymbol{y}^{(t)}) - \mathcal{L}(\boldsymbol{x}^{(t)}, \boldsymbol{y}^{(t-1)})$$
$$= -\mathcal{L}(\boldsymbol{x}^{(t)}, \boldsymbol{y}^{(t-1)}) - (-\mathcal{L}(\boldsymbol{x}^{(t)}, \boldsymbol{y}^{(t)}))$$
$$\leq (\frac{1}{n} A_{I^{(t)}, :} \boldsymbol{x}^{(t)} - \boldsymbol{g}_{I^{(t)}})^\top (\boldsymbol{y}_{I^{(t)}}^{(t)} - \boldsymbol{y}_{I^{(t)}}^{(t-1)}) + \frac{1}{2\alpha} \|\boldsymbol{y}_{I^{(t)}}^{(t-1)} - \boldsymbol{y}_{I^{(t)}}^{(t)}\|^2$$
$$= (\frac{1}{\delta} + \frac{1}{2\alpha}) \|\boldsymbol{y}^{(t)} - \boldsymbol{y}^{(t-1)}\|^2. \qquad (21)$$
Also, with the update rule of dual variables, we could make use of Eqn. (19) and re-write Eqn. (20) as:
$$\Delta_d^{(t)} - \Delta_d^{(t-1)}$$
$$\leq -\langle \frac{1}{n} A_{I^{(t)}, :} \bar{\boldsymbol{x}}^{(t)} - \boldsymbol{g}_{I^{(t)}}, \boldsymbol{y}_{I^{(t)}}^{(t)} - \boldsymbol{y}_{I^{(t)}}^{(t-1)} \rangle - \frac{1}{\delta} \|\boldsymbol{y}^{(t)} - \boldsymbol{y}^{(t-1)}\|^2$$
$$+ (\boldsymbol{y}^{(t)} - \boldsymbol{y}^{(t-1)})^\top (\frac{1}{n} A_{I^{(t)}, :} \boldsymbol{x}^{(t)} - \boldsymbol{g}_{I^{(t)}}) - \frac{1}{2\beta} \|\boldsymbol{y}^{(t)} - \boldsymbol{y}^{(t-1)}\|^2$$
$$= -\langle \frac{1}{n} A_{I^{(t)}, :} (\bar{\boldsymbol{x}}^{(t)} - \boldsymbol{x}^{(t)}), \boldsymbol{y}_{I^{(t)}}^{(t)} - \boldsymbol{y}_{I^{(t)}}^{(t-1)} \rangle - (\frac{1}{\delta} + \frac{1}{2\beta}) \|\boldsymbol{y}^{(t)} - \boldsymbol{y}^{(t-1)}\|^2 \qquad (22)$$

Together we get:

$$\Delta_d^{(t)} - \Delta_d^{(t-1)} + \Delta_p^{(t)} - \Delta_p^{(t-1)}$$

$$=\mathcal{L}(\boldsymbol{x}^{(t+1)}, \boldsymbol{y}^{(t)}) - \mathcal{L}(\boldsymbol{x}^{(t)}, \boldsymbol{y}^{(t)}) + \mathcal{L}(\boldsymbol{x}^{(t)}, \boldsymbol{y}^{(t)}) - \mathcal{L}(\boldsymbol{x}^{(t)}, \boldsymbol{y}^{(t-1)})$$
$$\quad + 2(\Delta_d^{(t)} - \Delta_d^{(t-1)})$$

$$\leq \mathcal{L}(\boldsymbol{x}^{(t+1)}, \boldsymbol{y}^{(t)}) - \mathcal{L}(\boldsymbol{x}^{(t)}, \boldsymbol{y}^{(t)}) + (\frac{1}{\delta} + \frac{1}{2\alpha})\|\boldsymbol{y}_{I^{(t)}}^{(t-1)} - \boldsymbol{y}_{I^{(t)}}^{(t)}\|^2 + 2(\Delta_d^{(t)} - \Delta_d^{(t-1)})$$

$$\text{(from Eqn. (21))}$$

$$\leq \mathcal{L}(\boldsymbol{x}^{(t+1)}, \boldsymbol{y}^{(t)}) - \mathcal{L}(\boldsymbol{x}^{(t)}, \boldsymbol{y}^{(t)}) + (\frac{1}{\delta} + \frac{1}{2\alpha})\|\boldsymbol{y}_{I^{(t)}}^{(t-1)} - \boldsymbol{y}_{I^{(t)}}^{(t)}\|^2$$
$$\quad - 2\langle \frac{1}{n}A_{I^{(t)},:}(\bar{\boldsymbol{x}}^{(t)} - \boldsymbol{x}^{(t)}), \boldsymbol{y}_{I^{(t)}}^{(t)} - \boldsymbol{y}_{I^{(t)}}^{(t-1)}\rangle - 2(\frac{1}{\delta} + \frac{1}{2\beta})\|\boldsymbol{y}^{(t)} - \boldsymbol{y}^{(t-1)}\|^2$$

$$\text{(from Eqn. (22))}$$

$$=\mathcal{L}(\boldsymbol{x}^{(t+1)}, \boldsymbol{y}^{(t)}) - \mathcal{L}(\boldsymbol{x}^{(t)}, \boldsymbol{y}^{(t)}) - 2\langle \frac{1}{n}A_{I^{(t)},:}(\bar{\boldsymbol{x}}^{(t)} - \boldsymbol{x}^{(t)}), \boldsymbol{y}_{I^{(t)}}^{(t)} - \boldsymbol{y}_{I^{(t)}}^{(t-1)}\rangle$$
$$\quad - (\frac{1}{\delta} + \frac{1}{\beta} - \frac{1}{2\alpha})\|\boldsymbol{y}^{(t)} - \boldsymbol{y}^{(t-1)}\|^2$$

$$\leq \mathcal{L}(\boldsymbol{x}^{(t+1)}, \boldsymbol{y}^{(t)}) - \mathcal{L}(\boldsymbol{x}^{(t)}, \boldsymbol{y}^{(t)}) + 2\delta\|\frac{1}{n}A_{I^{(t)},:}(\bar{\boldsymbol{x}}^{(t)} - \boldsymbol{x}^{(t)})\|^2$$
$$\quad - (\frac{1}{\delta} - \frac{1}{2\delta})\|\boldsymbol{y}^{(t)} - \boldsymbol{y}^{(t-1)}\|^2 \qquad \text{(since } 2ab \leq \gamma a^2 + 1/\gamma b^2)$$

$$\leq \mathcal{L}(\boldsymbol{x}^{(t+1)}, \boldsymbol{y}^{(t)}) - \mathcal{L}(\boldsymbol{x}^{(t)}, \boldsymbol{y}^{(t)}) - \frac{1}{2\delta}\|\boldsymbol{y}^{(t)} - \boldsymbol{y}^{(t-1)}\|^2$$
$$\quad + \frac{2\delta Rk}{n^2}\|\bar{\boldsymbol{x}}^{(t)} - \boldsymbol{x}^{(t)}\|^2$$

$$\square$$

Therefore we will connect the progress induced by $-\|\boldsymbol{y}^{(t)} - \boldsymbol{y}^{(t-1)}\|$ and dual gap $\Delta_d^{(t)}$ next.

## A.5 Dual progress

**Claim A.4.** *An $\alpha$-strongly convex function $f$ satisfies:*

$$f(\boldsymbol{x}) - f^* \leq \frac{1}{2\alpha}\|\nabla f(\boldsymbol{x})\|_2^2$$

This simply due to $f(\boldsymbol{x}) - f^* \leq \langle \nabla f(\boldsymbol{x}), \boldsymbol{x} - \bar{\boldsymbol{x}}\rangle - \frac{\alpha}{2}\|\boldsymbol{x} - \bar{\boldsymbol{x}}\|_2^2 \leq \frac{1}{2\alpha}\|\nabla f(\boldsymbol{x})\|^2 + \frac{\alpha}{2}\|\boldsymbol{x} - \bar{\boldsymbol{x}}\|^2 - \frac{\alpha}{2}\|\boldsymbol{x} - \bar{\boldsymbol{x}}\|^2 = \frac{1}{2\alpha}\|\nabla f(\boldsymbol{x})\|^2$.

Since $-D$ is $\frac{1}{\beta}$-strongly convex, we get

$$\Delta_d^{(t)} = D^* - D(\boldsymbol{y}^{(t)}) \leq \frac{\beta}{2}\|\nabla D(\boldsymbol{y}^{(t)})\|_2^2$$
$$= \frac{\beta}{2}\|\frac{1}{n}A\bar{\boldsymbol{x}}^{(t)} - \boldsymbol{g}\|_2^2$$
$$\leq \frac{n\beta}{2k}\|\frac{1}{n}A_{\bar{I},:}\bar{\boldsymbol{x}}^{(t)} - \boldsymbol{g}_{\bar{I}}\|_2^2, \qquad (23)$$

where $\bar{I}$ is a set of size $k$ that maximizes the values of $A_i^\top \bar{\boldsymbol{x}}^{(t)} - g_i$.

**Lemma A.5** (Dual Progress).

$$-\|\boldsymbol{y}^{(t)} - \boldsymbol{y}^{(t-1)}\|^2 \leq -\frac{k\delta}{n\beta}\Delta_d^{(t)} + \frac{k\delta}{n^2}R\|\bar{\boldsymbol{x}}^{(t)} - \boldsymbol{x}^{(t)}\|_2^2$$

*Proof of Lemma A.5.* Define $\Delta = \frac{1}{n}A(\bar{\boldsymbol{x}}^{(t)} - \boldsymbol{x}^{(t)})$. Since

$$- \|\frac{1}{n}A_{I^{(t)}}^\top \boldsymbol{x}^{(t)} - \boldsymbol{g}_{I^{(t)}}\|^2$$

$$\leq - \|\frac{1}{n}A_{\bar{I}}^\top \boldsymbol{x}^{(t)} - \boldsymbol{g}_{\bar{I}}\|^2 \qquad \text{(choice of } I^{(t)})$$

$$= - \|\frac{1}{n}A_{\bar{I}}^\top \bar{\boldsymbol{x}}^{(t)} - \boldsymbol{g}_{\bar{I}} - \Delta_{\bar{I}}\|^2$$

$$\leq - \frac{1}{2}\|\frac{1}{n}A_{\bar{I}}^\top \bar{\boldsymbol{x}}^{(t)} - \boldsymbol{g}_{\bar{I}}\|^2 + \|\Delta_{\bar{I}}\|_2^2$$

$$\text{(since } -(a+b)^2 \leq -1/2a^2 + b^2)$$

$$\leq - \frac{k}{n\beta}\Delta_d^{(t)} + \|\Delta_{\bar{I}}\|_2^2 \qquad \text{(from (23))}$$

$$\leq - \frac{k}{n\beta}\Delta_d^{(t)} + \frac{k}{n^2}R\|\bar{\boldsymbol{x}}^{(t)} - \boldsymbol{x}^{(t)}\|_2^2$$

With the relation between $\frac{1}{n}A_{I^{(t)}}^\top \boldsymbol{x}^{(t)} - \boldsymbol{g}_{I^{(t)}}$ and $\boldsymbol{y}^{(t)} - \boldsymbol{y}^{(t-1)}$ we finish the proof. $\qquad\square$

## A.6 Convergence on Duality Gap

Now we are able to merge the primal/dual progress to get the overall progress on the duality gap.

*Proof of Theorem 4.1.* We simply blend Lemma A.2 and Lemma A.5 with the primal-dual progress (Lemma A.3):

$$\Delta_d^{(t)} - \Delta_d^{(t-1)} + \Delta_p^{(t)} - \Delta_p^{(t-1)}$$

$$\leq \mathcal{L}(\boldsymbol{x}^{(t+1)}, \boldsymbol{y}^{(t)}) - \mathcal{L}(\boldsymbol{x}^{(t)}, \boldsymbol{y}^{(t)}) - \frac{1}{2\delta}\|\boldsymbol{y}^{(t)} - \boldsymbol{y}^{(t-1)}\|^2$$

$$+ \frac{2\delta Rk}{n^2}\|\bar{\boldsymbol{x}}^{(t)} - \boldsymbol{x}^{(t)}\|^2 \qquad \text{(Lemma A.3)}$$

$$\leq \mathcal{L}(\boldsymbol{x}^{(t+1)}, \boldsymbol{y}^{(t)}) - \mathcal{L}(\boldsymbol{x}^{(t)}, \boldsymbol{y}^{(t)}) + \frac{\delta}{2}(-\frac{k}{n\beta}\Delta_d^{(t)} + \frac{k}{n^2}R\|\bar{\boldsymbol{x}}^{(t)} - \boldsymbol{x}^{(t)}\|_2^2)$$

$$+ \frac{2\delta Rk}{n^2}\|\bar{\boldsymbol{x}}^{(t)} - \boldsymbol{x}^{(t)}\|^2 \qquad \text{(Lemma A.5)}$$

$$= \mathcal{L}(\boldsymbol{x}^{(t+1)}, \boldsymbol{y}^{(t)}) - \mathcal{L}(\boldsymbol{x}^{(t)}, \boldsymbol{y}^{(t)}) - \frac{k\delta}{2n\beta}\Delta_d^{(t)} + \frac{5R\delta k}{2n^2}\|\bar{\boldsymbol{x}}^{(t)} - \boldsymbol{x}^{(t)}\|_2^2$$

$$\leq \mathcal{L}(\boldsymbol{x}^{(t+1)}, \boldsymbol{y}^{(t)}) - \mathcal{L}(\boldsymbol{x}^{(t)}, \boldsymbol{y}^{(t)}) - \frac{k\delta}{2n\beta}\Delta_d^{(t)} + \frac{5R\delta k}{\mu n^2}(\mathcal{L}(\boldsymbol{x}^{(t)}, \boldsymbol{y}^{(t)}) - \mathcal{L}(\bar{\boldsymbol{x}}^{(t)}, \boldsymbol{y}^{(t)}))$$

$$= (1 - \frac{5R\delta k}{\mu n^2})(\mathcal{L}(\boldsymbol{x}^{(t+1)}, \boldsymbol{y}^{(t)}) - \mathcal{L}(\boldsymbol{x}^{(t)}, \boldsymbol{y}^{(t)})) - \frac{k\delta}{2n\beta}\Delta_d^{(t)}$$

$$+ \frac{5R\delta k}{\mu n^2}(\mathcal{L}(\boldsymbol{x}^{(t+1)}, \boldsymbol{y}^{(t)}) - \mathcal{L}(\bar{\boldsymbol{x}}^{(t)}, \boldsymbol{y}^{(t)}))$$

$$\leq - \frac{k\delta}{2n\beta}\Delta_d^{(t)} - \left((1 - \frac{5R\delta k}{\mu n^2})\frac{\mu}{4L} - \frac{5R\delta k}{\mu n^2}\right)\Delta_p^{(t)} \qquad \text{(Lemma A.2)}$$

When setting $\frac{k\delta}{2n\beta} = (1 - \frac{5R\delta k}{\mu n^2})\frac{\mu}{4L} - \frac{5R\delta k}{\mu n^2}$, we get that $\Delta^{(t)} \leq \frac{1}{1+a}\Delta^{(t-1)}$, where $1/a = \mathcal{O}(\frac{L}{\mu}(1 + \frac{R\beta}{n\mu}))$. Therefore it takes $\mathcal{O}(\frac{L}{\mu}(1 + \frac{R\beta}{n\mu})\log\frac{1}{\epsilon})$ for $\Delta^{(t)}$ to reach $\epsilon$.

When $\beta > 2\alpha$, we could redefine the primal-dual process as $\Delta^{(t)} := (\frac{\beta}{\alpha} - 1)\Delta_d^{(t)} + \Delta_p^{(t)}$ and rewrite some of the key steps, especially for the overall primal-dual progress.

$$\Delta^{(t)} - \Delta^{(t-1)}$$

$$=(\frac{\beta}{\alpha}-1)(\Delta_d^{(t)}-\Delta_d^{(t-1)})+\Delta_p^{(t)}-\Delta_p^{(t-1)}$$

$$=\mathcal{L}(\boldsymbol{x}^{(t+1)},\boldsymbol{y}^{(t)})-\mathcal{L}(\boldsymbol{x}^{(t)},\boldsymbol{y}^{(t)})+\mathcal{L}(\boldsymbol{x}^{(t)},\boldsymbol{y}^{(t)})-\mathcal{L}(\boldsymbol{x}^{(t)},\boldsymbol{y}^{(t-1)})$$
$$+\frac{\beta}{\alpha}(\Delta_d^{(t)}-\Delta_d^{(t-1)})$$

$$\leq\mathcal{L}(\boldsymbol{x}^{(t+1)},\boldsymbol{y}^{(t)})-\mathcal{L}(\boldsymbol{x}^{(t)},\boldsymbol{y}^{(t)})+(\frac{1}{\delta}+\frac{1}{2\alpha})\|\boldsymbol{y}_{I^{(t)}}^{(t-1)}-\boldsymbol{y}_{I^{(t)}}^{(t)}\|^2$$
$$-\frac{\beta}{\alpha}\langle\frac{1}{n}A_{I^{(t)},:}(\bar{\boldsymbol{x}}^{(t)}-\boldsymbol{x}^{(t)}),\boldsymbol{y}_{I^{(t)}}^{(t)}-\boldsymbol{y}_{I^{(t)}}^{(t-1)}\rangle-\frac{\beta}{\alpha}(\frac{1}{\delta}+\frac{1}{2\beta})\|\boldsymbol{y}^{(t)}-\boldsymbol{y}^{(t-1)}\|^2$$

<div align="right">(from Eqn. (21) and (22))</div>

$$=\mathcal{L}(\boldsymbol{x}^{(t+1)},\boldsymbol{y}^{(t)})-\mathcal{L}(\boldsymbol{x}^{(t)},\boldsymbol{y}^{(t)})-\frac{\beta}{\alpha}\langle\frac{1}{n}A_{I^{(t)},:}(\bar{\boldsymbol{x}}^{(t)}-\boldsymbol{x}^{(t)}),\boldsymbol{y}_{I^{(t)}}^{(t)}-\boldsymbol{y}_{I^{(t)}}^{(t-1)}\rangle$$
$$-(\frac{\beta}{\alpha}-1)\frac{1}{\delta}\|\boldsymbol{y}^{(t)}-\boldsymbol{y}^{(t-1)}\|^2$$

$$\leq\mathcal{L}(\boldsymbol{x}^{(t+1)},\boldsymbol{y}^{(t)})-\mathcal{L}(\boldsymbol{x}^{(t)},\boldsymbol{y}^{(t)})+\frac{3\beta}{2\alpha}\delta\|\frac{1}{n}A_{I^{(t)},:}(\bar{\boldsymbol{x}}^{(t)}-\boldsymbol{x}^{(t)})\|^2$$
$$-(\frac{3\beta}{4\alpha}-1)\frac{1}{\delta}\|\boldsymbol{y}^{(t)}-\boldsymbol{y}^{(t-1)}\|^2 \qquad\text{(since } ab\leq\delta a^2+1/(4\delta)b^2)$$

$$\leq\mathcal{L}(\boldsymbol{x}^{(t+1)},\boldsymbol{y}^{(t)})-\mathcal{L}(\boldsymbol{x}^{(t)},\boldsymbol{y}^{(t)})+\frac{\beta}{\alpha}\delta\|\frac{1}{n}A_{I^{(t)},:}(\bar{\boldsymbol{x}}^{(t)}-\boldsymbol{x}^{(t)})\|^2$$
$$-\frac{\beta}{4\alpha\delta}\|\boldsymbol{y}^{(t)}-\boldsymbol{y}^{(t-1)}\|^2 \qquad\text{(since } \beta/\alpha\geq 2)$$

$$\leq\mathcal{L}(\boldsymbol{x}^{(t+1)},\boldsymbol{y}^{(t)})-\mathcal{L}(\boldsymbol{x}^{(t)},\boldsymbol{y}^{(t)})-\frac{\beta}{4\alpha\delta}\|\boldsymbol{y}^{(t)}-\boldsymbol{y}^{(t-1)}\|^2$$
$$+\frac{\beta\delta Rk}{\alpha n^2}\|\bar{\boldsymbol{x}}^{(t)}-\boldsymbol{x}^{(t)}\|^2$$

Similarly to the previous setting, we get the whole primal-dual progress is bounded as follows:

$$(\frac{\beta}{\alpha}-1)(\Delta_d^{(t)}-\Delta_d^{(t-1)})+\Delta_p^{(t)}-\Delta_p^{(t-1)}$$

$$\leq\mathcal{L}(\boldsymbol{x}^{(t+1)},\boldsymbol{y}^{(t)})-\mathcal{L}(\boldsymbol{x}^{(t)},\boldsymbol{y}^{(t)})-\frac{\beta\delta}{4\alpha}\frac{k}{n\beta}\Delta_d^{(t)}$$
$$+\frac{5\beta R\delta k}{2\alpha\mu n^2}(\mathcal{L}(\boldsymbol{x}^{(t)},\boldsymbol{y}^{(t)})-\mathcal{L}(\bar{\boldsymbol{x}}^{(t)},\boldsymbol{y}^{(t)}))$$

$$\leq-\frac{\beta}{4\alpha}\frac{k\delta}{n\beta}\Delta_d^{(t)}-\left((1-\frac{5\beta R\delta k}{2\alpha\mu n^2})\frac{\mu}{4L}-\frac{5\beta R\delta k}{2\alpha\mu n^2}\right)\Delta_p^{(t)}$$

Therefore, when we set a proper $k$ and $\delta$ such that $\frac{\beta}{4\alpha}\frac{k\delta}{n\beta}=(\frac{\beta}{\alpha}-1)\left((1-\frac{5\beta R\delta k}{2\alpha\mu n^2})\frac{\mu}{4L}-\frac{5\beta R\delta k}{2\alpha\mu n^2}\right)$, and since $\frac{\beta}{\alpha}-1\geq\frac{\beta}{2\alpha}$, we get $\delta=\frac{1}{k}(\frac{L}{\mu n\beta}+\frac{5\beta R}{2\alpha\mu n^2}(1+4\frac{L}{\mu}))^{-1}$. And we have $\Delta^{(t)}-\Delta^{(t-1)}\leq-1/a\Delta^{(t)}$, where $a=\mathcal{O}(\frac{L}{\mu}(1+\frac{\beta}{\alpha}\frac{R\beta}{n\mu}))$. Therefore it takes $t=\mathcal{O}(\frac{L}{\mu}(1+\frac{\beta}{\alpha}\frac{R\beta}{n\mu})\log\frac{1}{\epsilon})$ iterations for the duality gap $\Delta^{(t)}$ to reach $\epsilon$ error. $\qquad\square$

## A.7 Smooth Hinge Loss and Relevant Properties

Smooth hinge loss is defined as follows:

$$h(z)=\begin{cases}\frac{1}{2}-z & \text{if } z<0\\\frac{1}{2}(1-z)^2 & \text{if } z\in[0,1]\\0 & \text{otherwise.}\end{cases} \tag{24}$$

Our loss function over a prediction $p$ associated with a label $\ell_i \in \{\pm 1\}$ will be $f_i(p) = h(p\ell_i)$. The derivative of smooth hinge loss $h$ is:

$$h'(z) = \begin{cases} -1 & \text{if } z < 0 \\ z - 1 & \text{if } z \in [0, 1] \\ 0 & \text{otherwise.} \end{cases} \tag{25}$$

Its convex conjugate is:

$$h^*(z^*) = \begin{cases} \frac{1}{2}(z^*)^2 + z^* & \text{if } z^* \in [-1, 0] \\ \infty & \text{otherwise.} \end{cases} \tag{26}$$

Notice since $f_i(p) = h(\ell_i p)$, $f_i^*(p) = h^*(p/\ell_i) = h^*(p\ell_i)$.

**Claim A.6.** *For a convex and $\beta$-smooth scalar function $f$, if it is $\alpha$ strongly convex over some convex set, and linear otherwise, then its conjugate function $f^*$ is $1/\beta$-strongly convex, and it is a $1/\alpha$-smooth function plus an indicator function over some interval $[a, b]$.*

*Proof.* To begin with, since $f''(x) \leq \beta, \forall x$, meaning $f$ is $\beta$-smooth, then with duality we have $f^*$ is $1/\beta$ strongly convex [16]. Secondly, since $f$ is $\alpha$ strongly convex over a convex set, meaning an interval for $\mathbb{R}$, therefore $f$ could only be linear on $(-\infty, a]$ or $[b, \infty)$, and is $\alpha$-strongly convex over the set $[a, b]$ (Here for simplicity $a < b$ could be $\pm\infty$). We denote $f'(-\infty) := \lim_{x \to -\infty} f'(x)$ and $f'(-\infty)$ likewise. It's easy to notice that $f'(-\infty) \leq f'(a) < f'(b) \leq f'(\infty)$ since $f$ is convex overall and strongly convex over $[a, b]$. Therefore $f(y) > f(a) + f'(a)(y - a)$ when $y > a$ and $f(y) = f(a) + f'(a)(y - a)$ when $y \leq a$.

Now since $f^*(x^*) \equiv \max_x \{x^* x - f(x)\}$, it's easy to verify that when $x^* < f'(a)$, $x^* x - f(x) = x^* x - f(a) - f'(a)(x - a) = -(f'(a) - x^*)x - f(a) + f'(a)a \to \infty$ when $x \to -\infty$. Similarly, when $x^* > f'(b)$, $f^*(x^*) = \infty$. On the other hand, when $x^* \in [f'(a), f'(b)]$, $f^*(x^*) = \max_x \{x^* x - f(x)\} = \max_{x \in [a,b]} \{x^* x - f(x)\}$. This is because $x^* a - f(a) \geq x^* y - f(y) = x^* y - f(y) - f'(a)(y - a), \forall y \leq a$, and similarly $x^* b - f(b) \geq x^* y - f(y) \forall y > b$. Therefore $f^*$ is $1/\alpha$ smooth over the interval $[f'(a), f'(b)]$, where $-\infty \leq f'(a) < f'(b) \leq \infty$. $\square$

### A.8 Convergence of Optimization over Trace Norm Ball

The convergence analysis for trace norm ball is mostly similar to the case of $\ell_1$ ball. The most difference lies on the primal part, where our approximated update incur linear progress as well as some error.

**Lemma A.7** (Primal Progress for Algorithm 2). *Suppose rank $\bar{X}^{(t)} \leq s$ and $\epsilon > 0$. If each $\tilde{X}$ computed in our algorithm is a $(\frac{1}{2}, \frac{\epsilon}{8})$-approximate solution to (18), then for every $t$, it satisfies $\mathcal{L}(X^{(t+1)}, Y^{(t)}) - \mathcal{L}(X^{(t)}, Y^{(t)}) \leq -\frac{\mu}{8L}\Delta_p^{(t)} + \frac{\epsilon}{16}$.*

*Proof.* Refer to the proof in [1] we have:

$$\mathcal{L}(X^{(t+1)}, Y^{(t)}) - \mathcal{L}(\bar{X}^{(t)}, Y^{(t)}) \leq (1 - \frac{\mu}{8L})\left(\mathcal{L}(X^{(t)}, Y^{(t)})) - \mathcal{L}(\bar{X}^{(t)}, Y^{(t)})\right) + \frac{\epsilon\mu}{16L}$$

Now move the first term on the RHS to the left and rearrange we get:

$$(1 - \frac{\mu}{8L})(\mathcal{L}(X^{(t+1)}, Y^{(t)}) - \mathcal{L}(X^{(t)}, Y^{(t)})) + \frac{\mu}{8L}\left(\mathcal{L}(X^{(t+1)}, Y^{(t)})) - \mathcal{L}(\bar{X}^{(t)}, Y^{(t)})\right) \leq \frac{\epsilon\mu}{16L}$$

Therefore we get:

$$\mathcal{L}(X^{(t+1)}, Y^{(t)}) - \mathcal{L}(X^{(t)}, Y^{(t)})) \leq -\frac{\mu}{8L}\Delta_p^{(t)} + \frac{\epsilon}{16}.$$
$\square$

Now back to the convergence guarantees on the trace norm ball.

*Proof of Theorem 4.3.* We again define $\Delta = \frac{1}{n}A(\bar{X}^{(t)} - X^{(t)})$. $G = \nabla_Y \mathcal{L}(X^{(t)}, Y^{(t)})$ such that $Y_{I^{(t)},:}^{(t)} - Y_{I^{(t)},:}^{(t-1)} = \delta(\frac{1}{n}\langle A_{I^{(t)},:}X^{(t)}\rangle - G_{I^{(t)},:})$. Again we get $\|\Delta\|_F^2 \leq \frac{R}{n^2}\|\bar{X}^{(t)} - X^{(t)}\|_F^2$.

$$\Delta_d^{(t)} \leq \frac{\beta}{2}\|\frac{1}{n}A\bar{X}^{(t)} - G\|_F^2 \leq \frac{n\beta}{2k}\|\frac{1}{n}A_{I^{(t)},:}\bar{X}^{(t)} - G_{I^{(t)},:}\|_F^2$$

Other parts are exactly the same and we get:

$$(\frac{\beta}{\alpha} - 1)(\Delta_d^{(t)} - \Delta_d^{(t-1)}) + \Delta_p^{(t)} - \Delta_p^{(t-1)}$$

$$\leq \mathcal{L}(X^{(t+1)}, Y^{(t)}) - \mathcal{L}(X^{(t)}, Y^{(t)}) - \frac{\beta\delta}{4\alpha}\frac{k}{n\beta}\Delta_d^{(t)}$$

$$+ \frac{5\beta R\delta k}{2\alpha\mu n^2}(\mathcal{L}(X^{(t)}, Y^{(t)}) - \mathcal{L}(\bar{X}^{(t)}, Y^{(t)}))$$

$$\leq -\frac{\beta}{4\alpha}\frac{k\delta}{n\beta}\Delta_d^{(t)} - \left((1 - \frac{5\beta R\delta k}{2\alpha\mu n^2})\frac{\mu}{8L} - \frac{5\beta R\delta k}{2\alpha\mu n^2}\right)\Delta_p^{(t)} + (1 - \frac{5\beta R\delta k}{2\alpha\mu n^2})\frac{\epsilon}{16}$$

(Lemma A.7)

Therefore when $\delta \leq \frac{1}{k}(\frac{L}{\mu n\beta} + \frac{5\beta R}{2\alpha\mu n^2}(1 + 8\frac{L}{\mu}))^{-1}$, it satisfies $\Delta^{(t)} - \Delta^{(t-1)} \leq -\frac{k\delta}{2\beta n}\Delta^{(t)} + \frac{\epsilon}{16}$. Therefore denote $a = \frac{2\beta n}{k\delta}$, we get $\Delta^{(t)} \leq \frac{a}{a+1}(\Delta^{(t-1)} + \frac{\epsilon}{16})$. Therefore we get $\Delta^{(t)} \leq (\frac{a}{a+1})^t\Delta^{(0)} + \frac{\epsilon}{16}\sum_{i=1}^{t}(\frac{a}{a+1})^i \leq (\frac{c}{c+1})^t\Delta^{(0)} + \epsilon/16$. Since $(\frac{a}{a+1})^t \leq e^{-t/a}$, it takes around $a = \mathcal{O}(\frac{L}{\mu}(1 + \frac{\beta}{\alpha}\frac{R\beta}{n\mu})\log\frac{1}{\epsilon})$ iterations for the duality gap to get $\epsilon$-error. $\qquad\square$

## A.9 Difficulty on Extension to Polytope Constraints

Another important type of constraint we have not explored in this paper is the polytope constraint. Specifically,

$$\min_{\boldsymbol{x} \in M \subset \mathbb{R}^d} f(A\boldsymbol{x}) + g(\boldsymbol{x}), M = conv(\mathcal{A}), \text{with only access to: } \text{LMO}_{\mathcal{A}(\boldsymbol{r})} \in \arg\min_{\boldsymbol{x} \in \mathcal{A}}\langle\boldsymbol{r}, \boldsymbol{x}\rangle,$$

where $\mathcal{A} \subset \mathbb{R}^d, |\mathcal{A}| = m$ is a finite set of vectors that is usually referred as atoms. It is worth noticing that this linear minimization oracle (LMO) for FW step naturally chooses a single vector in $\mathcal{A}$ that minimizes the inner product with $\boldsymbol{x}$. Again, this FW step creates some "partial update" that could be appreciated in many machine learning applications. Specifically, if our computation of gradient is again dominated by a matrix-vector (data matrix versus variable $\boldsymbol{x}$) inner product, we could possibly pre-compute each value of $\boldsymbol{v}_i := A\boldsymbol{x}_i, \boldsymbol{x}_i \in \mathcal{A}$, and simply use $\boldsymbol{v}_i$ to update the gradient information when $\boldsymbol{x}_i$ is the greedy direction provided by LMO.

When connecting to our sparse update case, we are now looking for a $k$-sparse update, $k \ll m = |\mathcal{A}|$, with the basis of $\mathcal{A}$, i.e., $\tilde{\boldsymbol{x}} = \sum_{i=1}^{k}\lambda_i\boldsymbol{x}_{n_i}, \boldsymbol{x}_{n_i} \in \mathcal{A}$. In this way, when we update $\boldsymbol{x}^+ \leftarrow (1 - \eta)\boldsymbol{x} + \eta\tilde{\boldsymbol{x}}$, we will only need to compute $\sum_{i=1}^{k}\boldsymbol{v}_{n_i}$ which is $\mathcal{O}(kd)$ time complexity.

However, to enforce such update that is "sparse" on $\mathcal{A}$ is much harder. To migrate our algorithms with $\ell_1$ ball or trace norm ball, we will essentially be solving the following problem:

$$\tilde{\boldsymbol{x}} \leftarrow \arg\min_{\Lambda \in \Delta^m, \|\Lambda\|_0 \leq k, \boldsymbol{x} = \sum_{i=1}^{m}\lambda_i\boldsymbol{x}_i, \boldsymbol{x}_i \in \mathcal{A}} \langle\boldsymbol{g}, \boldsymbol{y}\rangle + \frac{1}{2\eta}\|\boldsymbol{y} - \boldsymbol{x}\|_2^2,$$

where $\Delta^m$ is the $m$ dimensional simplex, and $\boldsymbol{g}$ is the current gradient vector.

Unlike the original sparse recovery problem that could be relaxed with an $\ell_1$ constraint to softly encourage sparsity, it's generally much harder to find the $k$ sparse $\Lambda$ in this case. Actually, it is as hard as the lattice problem [19] and is NP hard in general.

Therefore we are not able to achieve linear convergence with cheap update with polytope-type constraints. Nonetheless, the naive FW with primal dual formulation should still be computational efficient in terms of per iteration cost, where a concentration on SVM on its dual form has been explored by [22].

# B   Discussions on Efficient Coordinate Selections

The modified Block Frank-Wolfe step in Eqn. (3) achieves an $s$-sparse update of the iterates and could be computed efficiently when one knows which $s$ coordinates to update. However, in order to find the $s$ coordinates, one needs to compute the full gradient $\nabla f(\boldsymbol{x})$ with naive implementation. This phenomenon reminds us of greedy coordinate descent.

Even with the known fact that coordinate descent converges faster with greedy selection than with random order[30], there have been hardness to propogate this idea because of expensive greedy selections since the arguments that GCD converges similarly with RCD in [28], except for special cases [25, 24, 6, 17]. This is also probability why the partial updates nature of FW steps is less exploited before.

We investigate some possible tricks to boost GCD method that could be possibly applied to FW methods. A recent paper [17], Karimireddy et al. make connections between the efficient choice of the greedy coordinates with the problem of Maximum Inner Product Search (MIPS) for a composite function $P(\boldsymbol{x}) = f(A\boldsymbol{x}) + g(\boldsymbol{x})$, where $A \in \mathbb{R}^{n \times d}$. We rephrase the connection for the Frank-Wolfe algorithm. Since the computation of gradient is essentially $A^\top \nabla f_{|A\boldsymbol{x}} + \nabla g(\boldsymbol{x})$, to find its largest magnitude is to search maximum inner products among:

$$\pm \langle [\tilde{\boldsymbol{a}}_i^\top | 1], [\nabla f_{|A\boldsymbol{x}}^\top | \nabla_i g(\boldsymbol{x})] \rangle, \text{ i.e. } \pm \left( \tilde{\boldsymbol{a}}_i^\top \nabla f_{|A\boldsymbol{x}} + \nabla_i g(\boldsymbol{x}) \right),$$

where $\tilde{\boldsymbol{a}}_i \in \mathbb{R}^n$ is the $i$-th column of data matrix $A$, and $\nabla f_{|A\boldsymbol{x}}$ is the gradient of $f$ at $A\boldsymbol{x}$. In this way, we are able to select the greedy coordinates by conducting MIPS for a fixed $\mathbb{R}^{2d \times (n+1)}$ matrix $[A^\top | I | - A^\top | - I]^\top$ and each newly generated vector $[\nabla f_{|A\boldsymbol{x}}^\top | \nabla g_i(\boldsymbol{x})]$. Therefore when $\nabla g_i$ is constant for linear function or $\pm \lambda$ for $g(\boldsymbol{x}) = \lambda \|\boldsymbol{x}\|_1$, we could find the largest magnitude of the gradient in sublinear time. Still, the problems it could conquer is very limited. It doesn't even work for $\ell_2$ regularizer since the different coordinates in $\nabla_i g(\boldsymbol{x})$ creates $d$ new vectors in each iteration and traditional MIPS could resolve it in time sublinear to $d$. Meanwhile, even with constant $\nabla_i g(\boldsymbol{x})$, it still requires at least $\mathcal{O}((2d)^c \log(d))$ times of inner products of dimension $n + 1$ for some constant $c$ [34].

However, we have shown that for general composite form $f(A\boldsymbol{x}) + g(\boldsymbol{x})$ with much more relaxed requirements on the regularizer $g$, we are able to select and update each coordinate with *constant* times of inner products on average while achieving linear convergence. Therefore the usage of these tricks applied on FW method (MIPS as well as the nearest neighbor search [6]) is completely dominated by our contribution and we omit them in the main text of this paper.

# C   More Results on Empirical Studies

## C.1   More experiments with $\ell_1$ norm

To investigate more on how our algorithms perform with different choices of parameters, we conducted more empirical studies with different settings of condition numbers. Specifically, we vary the parameter $\mu$ that controls the strong convexity of the primal function. Experiments are shown in Figure 2.

## C.2   Experiments with trace norm ball on synthetic data

For trace norm constraints, we also implemented our proposal Primal Dual Block Frank Wolfe to compare with some prior work, especially Block FW [1]. Since prior work were mostly implemented in Matlab to tackle trace norm projections, we therefore also use Matlab to show fair comparisons. We choose quadratic loss $f(AX) = \|AX - B\|_F^2$ and $g$ to be $\ell_2$ regularizer with $\mu = 10/n$. The synthetic sensing matrix $A \in \mathbb{R}^{n \times d}$ is dense with $n = 1000$ and $d = 800$. Our observation $B$ is of dimension $1000 \times 600$ and is generated by a ground truth matrix $X_0$ such that $B = AX_0$. Here $X_0 \in \mathbb{R}^{800 \times 600}$ is constructed with low rank structure. We vary its rank $s$ to be $10, 20,$ and $100$. The comparisons with stochastic FW, blockFW [1], STORC [13], SCGS [23], and projected SVRG [15] are presented in Figure 3, which verifies that our proposal PDBFW consistently outperforms the baseline algorithms.

Figure 2: Convergence result comparison of different algorithms on smoothed hinge loss by varying the coefficient of the regularizer. The first row is the results ran on the rcv1.binary dataset, while the second row is the results ran on the news20.binary dataset. The first column is the result when the regularizer coeffcient $\mu$ is set to $1/n$. The middle column is when $\mu = 10/n$, and the right column is when $\mu = 100/n$.

Figure 3: Convergence comparison of our Primal Dual Block Frank Wolfe and other baselines. Figures show the relative primal objective value decreases with the wall time.