[Reviews · NeurIPS 2019]

Reviewer 1



== POST-REBUTTAL == Overall, I reckon that the authors have made a strong rebuttal. The contribution of the paper is now clearer in my mind and I reckon that both the theoretical result and experimental result of the paper look solid. As pointed out by other reviewers, I strongly encourage the authors to clarify the overall presentation based on our comments (highlight better the contributions, improve the presentation of experiments and algorithms...). For these reasons, I am ok to change my score to 7. == REVIEW BEFORE REBUTTAL == The paper is relatively well written and the proposed algorithms are technically sound. The experimental results on the 6 different datasets seem to indicate the performance of the approach. For these reasons I am leaning towards weak accept. However, as a disclaimer, I am not an expert in the domain and was not able to proofread all the proofs of the paper nor appreciate the complete significance of the contribution (hence my low confidence score). Below are some remarks/questions that could improve the paper: - Highlight better the contribution and significance of the paper: it would be important to add a paragraph in the introduction to highlight better why the proposed algorithm is a significant contribution for machine learning/optimization. - Equation (8) in Algorithm 1, sparse update: I was not able to understand exactly what was the procedure to obtain \tilde{x}. Do you have an exact algorithm? If yes can you give more details on that procedure (the section 7 in appendix was hard to understand). - How do you choose s in practice? - In the experimental section, please provide test accuracy curves as well. As this algorithm has a vocation to be used in Machine Learning one does not really care about getting to machine precision on the training set but we rather care about having good performance on the test set, hence it makes to also report performance on the test set. In particular, it is important to show that the algorithm is also better in the regimes of regularization that performs well in terms of final performance. - Related work: The authors should discuss the following related work: Frank-Wolfe algorithms for saddle point problems, Gauthier Gidel, Tony Jebara, Simon Lacoste-Julien - L92-93, for the three line proofs, I would still add details for the last line (using the \mu strong convexity) as you are in the constrained case and therefore the gradient at the optimum is not necessarily zero, this is not just a trivial application of the \mu strong convexity of f to me). - How does the method would compare to FW variants that also enjoy linear convergence rate (e.g. away FW (AFW) or pairwise FW)? In particular I think that it should be possible to apply the Block coordinate version of AFW presented in the following paper that also enjoys a linear convergence rate theoretically (under specific conditions) and in practice: Minding the Gaps for Block Frank-Wolfe Optimization of Structured SVMs, Osokin et al, ICML2016 It would be interesting to compare to that paper and discuss the advantages of your method compared to this work.

Reviewer 2



=== Update after reading rebuttal and reviews === We thank the authors for their detailed and convincing rebuttal. They have adequately addressed my concerns about both the correctness of the experiments, and the novelty of their theoretical analysis. However, I agree with reviewer 4 that the presentation of the results could be improved. In particular a discussion about DGPDC and details about the experiments should be added to the final version. Further, I agree with R4 that the title is misleading---it is not really a FW method, and is applicable in very specific settings (l1 and trace norm balls) and urge the authors to reconsider the current title. Given the above, I will increase my score from a 4 to a 6. # Summary This work deals with generalized least squares problems with a regularizer, and an additional constraint which can either be an l1 ball or a trace norm ball. The main result of the authors is a primal-dual algorithm where they show that if the optimal dual solution is always s-sparse, then all of the primal updates can be relaxed to also be s-sparse thereby resulting in significant savings. # Significance/Originality My main concern is that this work seems to nearly directly follow from [1]. The current results follow from combining the proof techniques from [1] with the observation from [2] for the trace norm constraint. Otherwise, the experimental results demonstrate a superior performance of the proposed method but I have some major concerns about their validity. # Concerns Major: 1. Comparison with [1] 2. It is never explicitly stated in the paper that the dual optimal is assumed to be sparse. This is quite a strong assumption and needs to be made more clear. 3. For the rcv1, news20, and mnist experiments (Fig 1), Acc projected gradient descent is faster than SVRG. This seems suspicious since typically the latter outperforms the former. I suspect this may be because multiple threads (i.e. parallelism) is being used which gives an unfair advantage to more expensive methods (including the proposed primal-dual FW). I would re-run the experiments with a single thread to have a fairer outcome and believe the current results are misleading. Minor 4. In Algorithms 1 and 2, step 6 involves greedily minimizing a quadratic function. This is no more a Frank-Wolfe method and is probably misleading to call it so. [1]: Lei, Q., Yen, I.E., Wu, C., Dhillon, I.S. & Ravikumar, P.. (2017). Doubly Greedy Primal-Dual Coordinate Descent for Sparse Empirical Risk Minimization. Proceedings of the 34th International Conference on Machine Learning, in PMLR 70:2034-2042 [2]: Allen-Zhu, Zeyuan, et al. "Linear convergence of a frank-wolfe type algorithm over trace-norm balls." Advances in Neural Information Processing Systems. 2017.

Reviewer 3



This paper proposes a novel method based on a combination of the Frank-Wolfe-like method in [1] (with an extension to sparsity problems) with a greedy coordinate ascent technique. Technical results are adequately sound: I did not encounter a major flaw in the analysis (note that I did not go thoroughly over the proofs in the supplements, but I did skim it). There are a few issues with the complexity analysis of the methods (listed at the end of this review), but these do not change the main advantage of the proposed method (that is avoiding the O(nd) cost of full matrix-vector multiplication with A). I would recommend emphasize, however, that these improved complexity guarantees come at the cost of knowing the sparsity (or rank) of the solution a priori. I am not particularly impressed by the clarity of this paper. It took me a few reads to understand some of the main contributions. One of the very first questions one would ask when reading the manuscript is "Why do we consider a primal-dual formulation while the template is completely smooth and strongly convex?". This should be clarified in an earlier stage of the paper. Some concerns: - Line 143: Reported O(d) and O(n) cost of quick selection holds only if you need to choose a single coordinate. As the proposed algorithm requires "s" and "k" coordinates, the costs should be O(d*log(s)) and O(n*log(k)) respectively, and this is using heap. - Line 144: It is not clear to me why the primal variable updates cost is O(s). A naive implementation requires O(d), as you also have a scaling factor (1-eta). - An extension of [1] for the sparse problems (but not to the primal-dual setup) should be included in the baseline methods in the numerical experiments. This would also clarify the significance of your approach. - Line 102: I guess one iteration of PGD costs O(nd) instead of O(d^2). - Line 135: How does "elastic net" fall into your template? It is a non-smooth regularizer (although strongly convex). Minor concerns: - The title is too generic, the template covered in the paper is kind of specific. It is restricted to ell-1 or trace norm-ball constraints. The optimization templates have strong convexity assumption on both terms. I would recommend using a more descriptive title. - Please consider defining problem parameters like "d" and "n" in the first equation where you introduce the problem template. I would also list the structural assumptions right here at the beginning, such as smoothness and the strong duality of "f" and "g", and that "C" is either ell-1 or trace norm-ball. This would be much more convenient for the readers. - Please avoid using contractions in lines 62 and 340 (doesn't -> does not) - Is "Delta x" in Eq(9) "x_tilde" ? - Eq (6): There are some Frank-Wolfe variants that can solve saddle point problems, although with sublinear rates. I would recommend the authors to have a look at [R1] and [R2, Section 5.4], as these methods might apply for solving (6). A completely subjective comment: I would not call this algorithm a Frank-Wolfe variant (of course the same goes for the algorithm developed in [1]), despite the similar per-iteration cost. The original Frank-Wolfe algorithm is a successive linear approximation approach. The main geometric intuition behind the original is minimizing a linear approximation of the function at the current estimate at each iteration. This in contrast with the classical first-order methods where we minimize a quadratic approximation. The method introduced in [1] (and used in this submission) minimizes a quadratic approximation with sparsity/rank constraint. This method does not recover the classical Frank-Wolfe algorithm even when we choose the sparsity/rank parameter 1. [R1] Gidel, Jebara, Lacoste-Julien, "Frank-Wolfe Algorithms for Saddle Point Problems", 2016. [R2] Yurtsever, Fercoq, Locatello, Cevher, "A Conditional Gradient Framework for Composite Convex Minimization with Applications to Semidefinite Programming", 2018. ========= After author feedback ========= I increase my score to 6 upon the author feedback and the discussions. I insist, however, that the manuscript requires a major update in terms of the presentation. I also respectfully disagree with the authors' feedback, that the proposed approach is the first primal-dual method for constrained problems. This is a vague claim that needs further specifications and clarifications. This is not even true for the line-up of Frank-Wolfe type algorithms. There is a mass of works that consider a primal-dual formulation of a constrained problem for various reasons. I write down a nonexhaustive list of examples: - Tran-Dinh, Cevher "A Primal-Dual Algorithmic Framework for Constrained Convex Minimization" - Liu, Liu, Ma, "On the Nonergodic Convergence Rate of an Inexact Augmented Lagrangian Framework for Composite Convex Programming" - Gidel, Pedregosa, Lacoste-Julien "Frank-Wolfe Splitting via Augmented Lagrangian Method" - Yurtsever, Fercoq, Cevher "A Conditional Gradient-Based Augmented Lagrangian Framework" - Silveti-Falls, Molinari, Fadili "Generalized Conditional Gradient with Augmented Lagrangian for Composite Minimization" ... I am not telling that these works are direct competitors for your approach, but these papers design and analyze primal-dual methods for constrained problems (or that also apply for constrained problems as a special case).

[Author Response · NeurIPS 2019]

We thank all the reviewers for their constructive comments and useful suggestions. Unfortunately there is significant
misunderstanding of our contributions. We will try to clarify here and also expand this in our paper.

**Q (R1, 4): Highlight our contributions:**
**A:** We proposed the first primal-dual algorithm for **constrained** problems. We are significantly more efficient compared
to the previous state of the art, both theoretically and empirically. Our method applies to a wide class of $\ell_1$ norm and
trace norm constrained problems including: ElasticNet, regularized SVMs and phase retrieval, among others. This is a
wide class of problems and a large body of prior optimization methods have been published. We have the most efficient
provable optimization method and this is a significant contribution.

**Q (R1, 4): How is the sparsity constraint chosen in practice? What if sparsity is underestimated?**
**A:** It's always safe to choose a relatively large target sparsity $s$. If one initially chooses a small $s$, one can increase it if
iterates converge but have smaller $\ell_1$ norm than the constrained value. If the sparsity is still underestimated, the sparsity
constrain dominated the $\ell_1$ constrain, and we will end up obtaining a solution with higher sparsity.

**Q (R1): Provide test accuracy to highlight effect of regularization**
**A:** We will add test accuracy as well as a comparison with different levels of regularization in the revised version.
However, our focus is on strongly convex objectives that guarantee a unique minimizer (same test error for different
algorithms), train accuracy has fully interpretation for the performance of the proposed algorithm compared among
others. Relevant literature commonly only reports train error in e.g. [1] or the DGPDC or BFW papers.

**Q (R1): How to obtain $\tilde{x}$ in equation 8?**
**A:** Equation 8 is a quadratic function about $\mathbf{x}$. Let's say $\tilde{\mathbf{x}} = \mathrm{argmin}_{\|\mathbf{x}\|_0 \leq s, \|x\|_1 \leq \lambda} \|\mathbf{x} - \mathbf{c}\|_2^2$. Then we obtain $\tilde{\mathbf{x}}$ by
performing an $\ell_0$ projection followed by an $\ell_1$ projection for $\mathbf{c}$.

**Q (R2): Why Accelerated Projected Gradient Descend (AccPGD) outperforms SVRG? Multi-threading?**
**A:** We implemented our algorithms as well as the baselines in C++ with the Eigen library, **without** multi-threading/multi-
processing. As for SVRG, please note that we are solving the constrained problem, and SVRG has to perform a
projection in every inner iteration. In the inner iteration of SVRG, the gradient computation is about $\mathcal{O}(sm)$ scaled by
$\mathrm{nnz}(A)/nd$ since our data is sparse, where $s$ is the sparsity during the inner step, and $m$ is the mini-batch size. While
projection on the $\ell_1$ ball requires $\mathcal{O}(d)$ (see Duchi et al. 2008). That is, projection on the $\ell_1$ ball can take more time
compared to the gradient computation. Empirically, projection takes about 75% of the CPU time of SVRG, and about
40% for AccPGD. As a side proof, the numerical results of [1] show that the performance of SVRG is not competitive.

**Q (R2, 4): Compared with Doubly Greedy Primal Dual Coordinate (DGPDC) and Block Frank Wolfe(BFW)?**
**A:** We are motivated from the primal dual reformulation like DGPDC and recent progress on FW like BFW, but our
new algorithm is not a trivial combination of previous results because: **1)** This is the first work to analyze constrained
problems using a primal-dual formulation. The challenges come from the non-symmetric formulation on primal and
dual variables. Prior work bounds the iteration progress and this will not work for our analysis. **2)** Besides, compared to
our results, the analysis of DGPDC highly relies on the **sparsity of the whole iterate trajectory**, which actually has
no obvious guarantee to be small. While our analysis only depends on **primal optimal's sparsity**, and is guaranteed
by the $\ell_1$ constraints. **3)** The inexact update in our algorithm 2 boosts the empirical performance, but also introduces
error in every iteration that perturbs the primal progress, and hence imposes more difficulties on the analysis under the
primal-dual framework.
Empirically we could not compared with DGPDC since it is not capable of solving constrained problems, but theoreti-
cally our sparsity requirement is more natural (on the primal optimal) rather than on the entire iterate trajectory. As for
BFW, both empirically and theoretically we have clearly demonstrated our improvements, when computing the full
gradient is expensive.

**Q (R2): The dual variable is assumed to be sparse?**
**A:** This is a very important point. We do not assume that the dual variables are sparse. In fact they will not be. Our
benefit is replacing the dimension $d$ to **primal** sparsity $s$. We will make sure this is more clear in the paper.

**Q (R4): Why primal dual formulation?**
**A:** The primal-dual reformulation ensures its gradient computation to be dominated by a bilinear term. Therefore, when
we compute the update with some (low-rank/sparse) structure, we are able to maintain the gradient and keep a cheap
update that is independent to the ambient dimension. For the primal framework, this only happens when the gradient is
linear in the update variable. This is clearly demonstrated in the theoretical vignette section (line 99 - 106).
We have also mentioned in the introduction (line 30 - 39): the primal-dual formulation allows us to exploit the sparsity
nature of the solution, and to reduce computational complexity from the ambient dimension to the solution's sparsity.

**Q (R4): Time complexity concerns and Elastic Net**
**A:** About quick select of complexity $\mathcal{O}(d)$, we choose the k-th largest value with linear operations and loop over the
coordinates to pick up all values greater than this value. Indeed the update operation is also $\mathcal{O}(d)$ or $\mathcal{O}(n)$ but it doesn't
affect the overall complexity. Line 104 comes from the warm-up problem where $f = 1/2\mathbf{x}^\top A\mathbf{x}$ and therefore PGD
costs $\mathcal{O}(d^2)$. As for elastic net we are refering to the constrained version on the $\ell_1$ regularization.

[1] Hazan, Elad, and Haipeng Luo."Variance-reduced and projection-free stochastic optimization." ICML 2016.


[Meta-Review · NeurIPS 2019]

After considering the rebuttal and discussing the paper, the reviewers have made a significant update to their scores, and came to a consensus to accept the paper, agreeing that it makes a nice contribution to NeurIPS, *assuming* the clarifications provided in the rebuttal are implemented in the camera ready version. It is important that the authors carefully update their camera ready given the reviewers comment (I will check!). Some side comments: - I fully agree with R4 that calling the method "Frank-Wolfe" is quite misleading. [1] had the excuse of considering a more specific problem (where the k-SVD can be argued to cost k 1-SVD). I suggest the authors rename the title to "Primal-Dual Block Generalized Frank-Wolfe", e.g., as generalized FW (e.g. "Generalized Conditional Gradient for Sparse Estimation", JMLR 2017) already exists with more powerful oracles than a LMO. - R1 and R4 mention some papers which should be mentioned in the related work. For example, for L64-66, [22] was extended in Osokin et al. ICML 2016 to obtain (almost) a linear convergence rate. Also, Gidel et al. AISTATS 2017 presents a FW method for saddle point problems which could be applied to (7) when the Fenchel dual function has compact support (e.g. with f_i being the hinge loss), and which has a linear convergence rate when the operator is strongly monotone.